# Outlier-Robust Distributionally Robust Optimization via Unbalanced Optimal Transport

**Zifan Wang**
KTH Royal Institute of Technology
zifanw@kth.se

**Yi Shen**
Duke University
yi.shen478@duke.edu

**Michael M. Zavlanos**
Duke University
michael.zavlanos@duke.edu

**Karl H. Johansson**
KTH Royal Institute of Technology
kallej@kth.se

## Abstract

Distributionally Robust Optimization (DRO) accounts for uncertainty in data distributions by optimizing the model performance against the worst possible distribution within an ambiguity set. In this paper, we propose a DRO framework that relies on a new distance inspired by Unbalanced Optimal Transport (UOT). The proposed UOT distance employs a soft penalization term instead of hard constraints, enabling the construction of an ambiguity set that is more resilient to outliers. Under smoothness conditions, we establish strong duality of the proposed DRO problem. Moreover, we introduce a computationally efficient Lagrangian penalty formulation for which we show that strong duality also holds. Finally, we provide empirical results that demonstrate that our method offers improved robustness to outliers and is computationally less demanding for regression and classification tasks.

## 1 Introduction

Consider a stochastic optimization problem that aims to find a decision variable $\theta \in \Theta$ that minimizes the expected loss function $\mathbb{E}_{\xi \sim \mathbb{P}^*}[l(\xi, \theta)]$ [1], where the random samples $\xi \in \Xi$ are drawn from a fixed distribution $\mathbb{P}^*$ and $\Theta \subset \mathbb{R}^d$ is a compact and convex set. In many applications, the distribution of interest $\mathbb{P}^*$ is not precisely known; yet datasets that contain finite samples independently drawn from the distribution $\mathbb{P}^*$ are often available, which yield an empirical distribution estimate $\hat{\mathbb{P}}$. For instance, in image classification problems, $\mathbb{P}^*$ can represent the joint distribution of the features and the labels, and $\hat{\mathbb{P}}$ can represent the empirical distribution of a dataset that is annotated by humans [2]; in portfolio optimization problems, $\mathbb{P}^*$ captures the uncertainty of a set of financial products in the future months and $\hat{\mathbb{P}}$ contains historical prices of all the products [3]; in healthcare applications, $\mathbb{P}^*$ can represent the feature distribution of a population of interest, and $\hat{\mathbb{P}}$ represent the available electronic health record data [4].

The above stochastic optimization problem is often solved using empirical risk minimization (ERM) [5], which searches for the best decision variable $\theta$ that minimizes the expected loss function with respect to the empirical distribution, i.e., $\theta_{\text{ERM}} = \arg\min_{\theta \in \Theta} \mathbb{E}_{\xi \sim \hat{\mathbb{P}}}[l(\xi, \theta)]$. However, ERM is not guaranteed to provide good out-of-sample performance, i.e., the value of $\mathbb{E}_{\xi \sim \mathbb{P}^*}[l(\xi, \theta_{\text{ERM}})]$ is high when the empirical estimate $\hat{\mathbb{P}}$ is not close to the true distribution $\mathbb{P}^*$. For example, in the presence of distribution shifts or contaminated datasets, the out-of-sample performance cannot be improved by increasing the sample size. To enhance the out-of-sample performance of the decision variable, researchers have developed methods that use distributionally robust optimization (DRO) to address

38th Conference on Neural Information Processing Systems (NeurIPS 2024).

the mismatch between $\hat{\mathbb{P}}$ and $\mathbb{P}^*$ [6, 7, 8, 9, 10, 11, 12]. In contrast to ERM, DRO aims to optimize outcomes against the worst-case scenario over a set of distributions that are close to the empirical distribution according to a pre-specified distance metric. This set of distributions is called the ambiguity set and is typically assumed to contain the distribution of interest, e.g., the true distribution $\mathbb{P}^*$. The construction of the ambiguity set is critical in the development of various DRO methods, as it ideally encompasses all prior knowledge about the true distribution. For example, [6] defines the ambiguity set using moment constraints. Moment constraints assume that the true distribution has similar moments compared to the empirical distribution, e.g., means and variances. The ambiguity set can also include distributions that are similar to $\hat{\mathbb{P}}$ in infinite-order moments; this construction is explored in [13] and is termed Kernel DRO. When the distribution of interest is categorical, Kullback-Leibler divergence (KL) is often adopted to define the ambiguity set as the categories can be permuted and KL does not consider the geometry of a distribution's support. DRO using KL has been extensively studied in the literature due to its computational benefits [14, 15]. In contrast, the Wasserstein distance between two distributions captures the geometry of the distributions. The Wasserstein distance can be useful, e.g., in loan approval prediction problems where the distribution represents the incomes of a population and prior knowledge may specify that two clients are similar if their incomes are close. DRO using the Wasserstein distance (WDRO) has been applied to many machine learning problems, such as linear regression [16] and logistic regression [17] problems.

In this paper, we apply DRO to handle distribution mismatches using the Wasserstein distance as it can better capture the geometry of the distributions. However, WDRO is not suitable for contaminated datasets, e.g., datasets that contain outliers that are geometrically far from the clean distribution but cannot be easily removed. The reason is that the Wasserstein distance between the contaminated empirical distribution and the clean (true) distribution is very sensitive to the outliers; to include the clean distribution in the ambiguity set, the Wasserstein distance would need to be selected very large and inevitably the ambiguity set would include distributions that will never happen in practice.

To apply DRO to contaminated datasets, the authors in [18, 19, 20] take outliers into consideration and propose various methods to minimize their impact on the learning performance. Specifically, [18] proposes to minimize the most optimistic DRO risk to avoid the hardest instances that are likely to be outliers. In [19], a weight clipping method is proposed to truncate the excessive impact of outliers on the results. However, both [18] and [19] use KL divergence to define the ambiguity set, which fails to capture geometric uncertainty. Notably, the authors in [20] introduce an outlier-robust WDRO framework capable of addressing both geometric uncertainty and non-geometric contamination, allowing an $\varepsilon$ fraction of data to be arbitrarily corrupted. They design an ambiguity set based on prior knowledge and demonstrate the strong duality of the resulting DRO problem, which is related to the Conditional Value at Risk (CVaR) [21]. Nevertheless, the performance of the formulated DRO problem relies heavily on the accuracy of the estimate of $\varepsilon$. Besides, [20] uses off-the-shelf solvers to solve the resulting DRO problem, which is not computationally efficient when applied to large datasets since the number of constraints in the proposed method grows linearly with the sample size.

The main contribution of this paper is to introduce a new framework for outlier-robust WDRO based on Unbalanced Optimal Transport (UOT), which is known for its inherent robustness to outliers and missing data [22, 23, 24, 25, 26]. Our approach involves the design of a UOT distance that substitutes hard constraints by a soft penalization term. This construction, combined with some prior knowledge on the learning task, enables the design of a new ambiguity set that includes distributions of interest and penalizes distributions that contain outliers. For the DRO problem with this new ambiguity set, we establish strong duality results under specific smoothness assumptions. However, the solution of the dual problem poses significant computational challenges. Motivated by [27], we explore a Lagrangian penalty variation of the problem, which can be viewed as a Lagrangian function of the original formulation with a fixed dual variable. We show that strong duality holds for this Lagrangian penalty problem under fewer assumptions compared to the original DRO problem. The reformulated problem employs an exponential function to reweight data points, effectively diminishing the influence of outlier data on the optimization process. We solve this problem by proposing a provable stochastic (sub)-gradient algorithm, which is computationally efficient. We provide empirical results that demonstrate that our approach not only enhances robustness to outliers but also improves computational efficiency for large-scale problems.

Our work is related to data reweighted optimization problems [28, 29, 30], which adjust the weights of individual data to manage outliers. However, these studies construct the reweighted optimization problems heuristically and cannot handle distribution uncertainty. In contrast, our problem

formulation is able to handle distribution uncertainty via DRO, and our findings demonstrate that the reweighted problem constitutes the dual of a specific class of DRO problems, with the ambiguity set defined by the UOT distance.

The rest of the paper is organized as follows. Section 2 defines the UOT distance and the associated DRO problem. Section 3 presents the strong duality results for both the original DRO problem and its Lagrangian penalty variant. The main algorithm along with its convergence analysis is presented in Section 4. Section 5 demonstrates the effectiveness of our proposed framework in addressing outliers in both regression and classification tasks. Finally, we conclude the paper in Section 6. In the Appendix, we provide all proofs as well as some additional experiments.

## 2    Problem definition and preliminaries

The (1-)Wasserstein distance [31] between two distributions $\mathbb{P}$ and $\hat{\mathbb{P}}$ is defined as

$$\mathrm{W}(\mathbb{P}, \hat{\mathbb{P}}) = \inf_{\gamma \in \Gamma(\mathbb{P}, \hat{\mathbb{P}})} \mathbb{E}_{(\xi, \zeta) \sim \gamma}[c(\xi, \zeta)], \tag{1}$$

where $\Gamma(\mathbb{P}, \hat{\mathbb{P}})$ denotes the set of joint distributions such that $\gamma_1 = \mathbb{P}$ and $\gamma_2 = \hat{\mathbb{P}}$, where $\gamma_1$ and $\gamma_2$ denote the first and the second marginal distribution of $\gamma$, respectively. The function $c(\xi, \zeta) : \Xi \times \Xi \to [0, +\infty)$ is lower semi-continuous and represents the cost of moving a point from $\xi$ to $\zeta$, where the support set $\Xi \subset \mathbb{R}^d$ is assumed to be compact. The joint distribution $\gamma$ specifies a transport plan for moving the distribution from $\mathbb{P}$ to $\hat{\mathbb{P}}$. The Wasserstein distance captures the underlying geometry in distributions through the cost function $c(\cdot, \cdot)$, making it a popular choice in DRO problems; the WDRO problem is defined by

$$\min_{\theta \in \Theta} \sup_{\mathbb{P} \in \mathcal{B}_\rho(\hat{\mathbb{P}})} \mathbb{E}_{\xi \sim \mathbb{P}}[l(\theta, \xi)], \tag{2}$$

where $\mathcal{B}_\rho(\hat{\mathbb{P}}) := \{\mathbb{P} \in \mathcal{M}(\Xi) : \mathrm{W}(\mathbb{P}, \hat{\mathbb{P}}) \leq \rho\}$ denotes the ambiguity set that contains all distributions within $\rho$-distance from $\hat{\mathbb{P}}$ and $\rho \geq 0$ is a user-specified parameter. Here, we denote by $\mathcal{M}(\Xi)$ the space of all probability distributions supported on $\Xi$.

The Wasserstein distance enforces equality constraints on the marginal distributions of $\gamma$ to match $\mathbb{P}$ and $\hat{\mathbb{P}}$, thereby restricting the choice of $\gamma$. If the empirical distribution $\hat{\mathbb{P}}$ contains outliers, it becomes necessary to select a large radius $\rho$ for the ambiguity set to ensure the inclusion of the true distribution $\mathbb{P}^*$ as the outliers tend to increase the Wasserstein distance between $\hat{\mathbb{P}}$ and $\mathbb{P}^*$. However, a large $\rho$ can lead to a DRO solution that is conservative since the larger the radius the larger the value of the inner supremum problem in (2). Though a large radius allows the inclusion of the distribution of interest, i.e., $\mathbb{P}^*$, it also results in the inclusion of many distributions that are not of interests, e.g., distributions that are unlikely to occur in practice. As a result, the Wasserstein distance is not a desired distance choice when the empirical distribution contains outliers.

In Unbalanced Optimal Transport (UOT) problems [22], the joint distribution $\gamma$ is not required to have fixed marginals in contrast to (1). Specifically, for any two arbitrary positive measures $\mathbb{P}$ and $\hat{\mathbb{P}}$, the UOT distance is defined by

$$\mathrm{UW}(\mathbb{P}, \hat{\mathbb{P}}) = \inf_{\gamma \geq 0} \mathbb{E}_{(\xi, \zeta) \sim \gamma}[c(\xi, \zeta)] + D_{\varphi_1}(\gamma_1 | \mathbb{P}) + D_{\varphi_2}(\gamma_2 | \hat{\mathbb{P}}), \tag{3}$$

where $\gamma_1$ and $\gamma_2$ are marginals of $\gamma$, and $D_{\varphi_1}$ and $D_{\varphi_2}$ are Csiszár divergences that measure discrepancies between positive measures, based on functions $\varphi_1$ and $\varphi_2$, respectively. The key distinction between UOT distance and Wasserstein (OT) distance lies in the constraints of the marginals. While the OT distance in (1) strictly enforces marginal distributions, i.e., $\gamma_1 = \mathbb{P}$ and $\gamma_2 = \hat{\mathbb{P}}$, the UOT in (3) relaxes the equality constraints by adding mismatch penalty functions $D_\varphi$ in addition to the transport cost induced by $c(\cdot, \cdot)$. This relaxation makes UOT robust to outliers since $\gamma_2$ can represent a distribution that is different from the contaminated distribution $\hat{\mathbb{P}}$.

Inspired by the definition of UOT, we consider the following unbalanced Wasserstein distance (divergence)

$$\mathrm{UW}(\mathbb{P}||\hat{\mathbb{P}}) = \inf_{\bar{\mathbb{P}}, \gamma \in \Gamma(\mathbb{P}, \bar{\mathbb{P}})} \left\{ \mathbb{E}_{(\xi, \zeta) \sim \gamma}[c(\xi, \zeta)] + \beta D_{KL}(\bar{\mathbb{P}}||\hat{\mathbb{P}}) \right\}, \tag{4}$$

where $\beta$ is a tuning parameter and $\bar{\mathbb{P}}$ is an intermediate optimization variable that links $\mathbb{P}$ and $\hat{\mathbb{P}}$. Note that (4) is a special case of (3) by selecting $\varphi_1 = \iota_{\{1\}}$ (i.e., $\varphi_1(1) = 0$ and $\infty$ otherwise) and $\varphi_2(x) = x \log x - x + 1$ (i.e., $D_{\varphi_2}$ denotes the KL divergence). In addition, we restrict the marginals of the joint positive measure $\gamma$ in (3) to be probability measures because the general positive measures are not of interest in the DRO problems we consider. The particular choice of $\varphi$ functions is for technical reasons related to the duality analysis, as we will show in Section 3.

The optimal unbalanced Wasserstein transport under $\mathrm{UW}(\mathbb{P}||\hat{\mathbb{P}})$ can be interpreted as a two-step procedure: 1) $\hat{\mathbb{P}}$ can be transported to any $\bar{\mathbb{P}}$ by paying a small price measured by its distance to $\hat{\mathbb{P}}$ in the KL sense (a non-geometric transport), and 2) $\bar{\mathbb{P}}$ is then transported to $\mathbb{P}$ by minimizing the geometry-aware transport cost. The non-geometric KL divergence is the key to allow the uncontaminated clean distribution $\mathbb{P}^*$ to be close to the contaminated empirical distribution $\hat{\mathbb{P}}$ as the KL divergence does not distinguish the geometric locations of the distribution supports.

We define the ambiguity set using the unbalanced Wasserstein distance by $\mathcal{U}_\rho(\hat{\mathbb{P}}) = \{\mathbb{P} \in \mathcal{M}(\Xi) : \mathrm{UW}(\mathbb{P}||\hat{\mathbb{P}}) \leq \rho\}$. Then, the DRO problem using the UOT distance (4) can be formulated as

$$\min_{\theta \in \Theta} \sup_{\mathbb{P} \in \mathcal{M}(\Xi) : \mathrm{UW}(\mathbb{P}||\hat{\mathbb{P}}) \leq \rho} \mathbb{E}_{\xi \sim \mathbb{P}}[l(\theta, \xi)]. \tag{5}$$

The ambiguity set defined by the unbalanced Wasserstein distance enables us to include the clean distribution $\mathbb{P}^*$, but it cannot remove distributions that are close to the contaminated empirical distribution $\hat{\mathbb{P}}$ as these distributions are close to $\hat{\mathbb{P}}$ both geometrically (Wasserstein) and non-geometrically (KL). As a result, these distributions, denoted by set $\tilde{\mathcal{P}}$, could also make the learned model conservative. To remove such distributions, we need some prior/domain knowledge. As discussed in [18, 20], it is impossible to design a model selection strategy using contaminated datasets without any prior knowledge. In Section 5, we also show that constructing the ambiguity set without prior knowledge may result in worse performance compared to the traditional DRO; see Table 5.

We assume that any distribution $\tilde{\mathbb{P}} \in \tilde{\mathcal{P}}$ incurs significantly large costs when evaluated using a given function $h$ compared to uncontaminated distributions, i.e., $\mathbb{E}_{\xi \sim \tilde{\mathbb{P}}}[h(\xi)] \gg \mathbb{E}_{\xi \sim \mathbb{P}}[h(\xi)], \forall \tilde{\mathbb{P}} \in \tilde{\mathcal{P}}$ and $\forall \mathbb{P} \in \{\mathbb{P} : \mathrm{UW}(\mathbb{P}||\hat{\mathbb{P}}) \leq \rho\} \backslash \tilde{\mathcal{P}}$. The function $h$ can be related to the moment constraint. For example, if we know the expectation estimate $\xi_0$ of the clean distribution, we can design $h(\xi) = \|\xi - \xi_0\|$ and the function value of $h(\xi)$ will be high if $\xi$ is an outlier. Similar choices are considered in [20].

Taking into account the prior knowledge, we consider minimizing the following (primal) function

$$(\mathrm{P}) = \sup_{\substack{\mathbb{P} \in \mathcal{M}(\Xi): \\ \mathrm{UW}(\mathbb{P}||\hat{\mathbb{P}}) \leq \rho}} \mathbb{E}_{\xi \sim \mathbb{P}}[l(\theta, \xi)] - \mathbb{E}_{\xi \sim \mathbb{P}}[h(\xi)] := \sup_{\substack{\mathbb{P} \in \mathcal{M}(\Xi): \\ \mathrm{UW}(\mathbb{P}||\hat{\mathbb{P}}) \leq \rho}} \mathbb{E}_{\xi \sim \mathbb{P}}[L(\theta, \xi)], \tag{6}$$

with $L(\theta, \xi) = l(\theta, \xi) - h(\xi)$ and we assume that $L(\theta, \xi)$ is continuous in $\xi$. For any distribution $\tilde{\mathbb{P}}$ that contains outliers, $\mathbb{E}_{\xi \sim \tilde{\mathbb{P}}}[h(\xi)]$ is large and thus $\tilde{\mathbb{P}}$ is less likely to attain the supremum in (6).

In summary, we first employ the unbalanced Wasserstein distance to incorporate the distributions of interest. Then, we leverage prior knowledge to systematically exclude distributions that may contain outliers. This two-step approach ensures a more thorough consideration of potential distributions while effectively mitigating the impact of outliers on the learning problem.

## 3 Unbalanced distributionally robust optimization

Note that the primal problem in (6) is an infinite-dimensional optimization as the optimization variable represents probability distributions. In this section, we derive its dual problem and a variant of the dual problem that is computationally tractable.

### 3.1 The DRO problem

**Assumption 1.** *Every joint distribution $\gamma$ on $\Xi \times \Xi$ with the second marginal distribution $\bar{\mathbb{P}}$ has a regular conditional distribution $\gamma_\zeta$ given the value of the second marginal equals $\zeta$.*

This assumption ensures that the joint distribution $\gamma$ can be written as $d\gamma(\xi, \zeta) = d\gamma_\zeta(\xi)d\bar{\mathbb{P}}(\zeta)$; see [32] for more details on regular conditional distributions. By the law of total expectation, we have $\mathbb{E}_{\xi \sim \mathbb{P}}[L(\theta, \xi)] = \mathbb{E}_{\zeta \sim \bar{\mathbb{P}}}\mathbb{E}_{\xi \sim \gamma_\zeta(\xi)}[L(\theta, \xi)]$.

We define $f_\lambda(\theta, \zeta) := \sup_{\xi \in \Xi}\{L(\theta, \xi) - \lambda c(\xi, \zeta)\}$. We first derive the dual problem and show the weak duality in the following lemma.

**Lemma 1.** *Let Assumption 1 hold and suppose that $|f_\lambda(\theta, \zeta)| < \infty$ for $\hat{\mathbb{P}}$-almost every $\zeta$. Define the dual problem $(D) := \inf_{\lambda \geq 0}\left\{\lambda\rho + \lambda\beta\log\mathbb{E}_{\zeta \sim \hat{\mathbb{P}}}\left[\exp\left(\frac{f_\lambda(\theta, \zeta)}{\lambda\beta}\right)\right]\right\}$. Then, we have*

$$(P) = \sup_{\mathbb{P} \in \mathcal{M}(\Xi):\mathrm{UW}(\mathbb{P}||\hat{\mathbb{P}}) \leq \rho} \mathbb{E}_{\xi \sim \mathbb{P}}[L(\theta, \xi)] \leq \inf_{\lambda \geq 0}\left\{\lambda\rho + \lambda\beta\log\mathbb{E}_{\zeta \sim \hat{\mathbb{P}}}\left[\exp\left(\frac{f_\lambda(\theta, \zeta)}{\lambda\beta}\right)\right]\right\} = (D).$$

The proof is provided in Appendix 7.1. To show the strong duality, we need two additional assumptions to guarantee that the function $f_\lambda(\theta, \zeta)$ is well-behaved.

**Assumption 2.** *The function $L(\theta, \xi)$ be differentiable and concave in $\xi$ with respect to the norm $\|\cdot\|$.*

**Assumption 3.** *The function $c : \Xi \times \Xi \to [0, \infty)$ is differentiable and $c(\cdot, \xi_0)$ is 1-strongly convex for each $\xi_0 \in \Xi$.*

**Theorem 1.** *Let Assumptions 1–3 hold and assume that $|f_\lambda(\theta, \zeta)| < \infty$ for $\hat{\mathbb{P}}$-almost every $\zeta$. Suppose that the optimal dual variable $\lambda^*$ is strictly positive. Then, the strong duality holds, i.e.,*

$$\sup_{\mathbb{P} \in \mathcal{M}(\Xi):\mathrm{UW}(\mathbb{P}||\hat{\mathbb{P}}) \leq \rho} \mathbb{E}_{\xi \sim \mathbb{P}}[L(\theta, \xi)] = \inf_{\lambda \geq 0}\left\{\lambda\rho + \lambda\beta\log\mathbb{E}_{\zeta \sim \hat{\mathbb{P}}}\left[\exp\left(f_\lambda(\theta, \zeta)/(\lambda\beta)\right)\right]\right\}. \quad (7)$$

The proof can be found in Appendix 7.2. For technical reasons, we show the strong duality when the optimal dual variable $\lambda^*$ is strictly positive. In most applications, the optimal dual variable $\lambda^*$ is in fact positive. When $\lambda^* = 0$, the inequality constraint becomes inactive and the radius of the ambiguity set becomes very large, a situation that is out of scope for most practical applications.

Although strong duality holds and the dual problem is a finite dimensional minimization problem, solving problem (7) is computationally challenging due to the complex dependency of the objective on $\lambda$ and, in particular, the implicit dependency of $f_\lambda$ on $\lambda$. Recall that $f_\lambda(\theta, \zeta) = \sup_{\xi \in \Xi}\{L(\theta, \xi) - \lambda c(\xi, \zeta)\}$; in general, we cannot obtain an analytical form of $f_\lambda(\theta, \zeta)$, i.e., a closed-form solution of the problem $\sup_{\xi \in \Xi}\{L(\theta, \xi) - \lambda c(\xi, \zeta)\}$. Additionally, the dual problem (7) is not convex in $\lambda$ in general. The task becomes more involved when attempting to minimize over $\lambda$ and $\theta$ jointly. To address these challenges, we shift focus from the constrained $\rho$-robustness problem (7) and instead consider the corresponding Lagrangian penalty problem, which allows for a computational tractable formulation.

### 3.2 The Lagrangian penalty problem

Motivated by [27], we consider the Lagrangian penalty reformulation of (7) as follows

$$\sup_{\mathbb{P} \in \mathcal{M}(\Xi)}\left\{\mathbb{E}_{\xi \sim \mathbb{P}}[L(\theta, \xi)] - \lambda\mathrm{UW}(\mathbb{P}||\hat{\mathbb{P}})\right\}. \quad (8)$$

In what follows, we show that the strong duality also holds for the Lagrangian penalty problem (8).

**Theorem 2.** *Let Assumption 1 hold and assume that $|f_\lambda(\theta, \zeta)| < \infty$ for $\hat{\mathbb{P}}$-almost every $\zeta$. Then, we have that*

$$\sup_{\mathbb{P} \in \mathcal{M}(\Xi)}\left\{\mathbb{E}_{\xi \sim \mathbb{P}}[L(\theta, \xi)] - \lambda\mathrm{UW}(\mathbb{P}||\hat{\mathbb{P}})\right\} = \lambda\beta\log\mathbb{E}_{\zeta \sim \hat{\mathbb{P}}}\left[\exp\left(f_\lambda(\theta, \zeta)/(\lambda\beta)\right)\right]. \quad (9)$$

The proof is provided in Appendix 7.3. Note that the strong duality of the Lagrangian penalty problem only requires Assumption 1 to hold in contrast to Theorem 1. In particular, it does not require the loss function $L(\theta, \xi)$ to be differentiable, making it applicable to a broader class of problems. Besides, solving problem (8) is computationally tractable as the dual problem has a closed-form solution as in (9). In particular, $\lambda$ is a parameter in (9) while it is an optimization (dual) variable in (7).

Since the logarithm function is monotonically increasing, minimization of (9) in $\theta$ is equivalent to the following form

$$\min_{\theta \in \Theta} \mathbb{E}_{\zeta \sim \hat{\mathbb{P}}} \left[ \exp \left( \sup_{\xi \in \Xi} \{ l(\theta, \xi) - h(\xi) - \lambda c(\xi, \zeta) \} / (\lambda \beta) \right) \right]. \tag{10}$$

**Comparison with standard WDRO.** It has been shown in [27] that the Lagrangian penalty problem using the Wasserstein metric $W_c$ has the following strong duality result

$$\min_{\theta \in \Theta} \left\{ \sup_{\mathbb{P} \in \mathcal{M}(\Xi)} \mathbb{E}_{\xi \sim \mathbb{P}}[l(\theta, \xi)] - \lambda W_c(\mathbb{P}, \hat{\mathbb{P}}) \right\} = \min_{\theta \in \Theta} \mathbb{E}_{\zeta \sim \hat{\mathbb{P}}} \left[ \sup_{\xi \in \Xi} \{ l(\theta, \xi) - \lambda c(\xi, \zeta) \} \right], \tag{11}$$

where $W_c$ is the Wasserstein distance metric associated with the cost function $c(\cdot, \cdot)$. When $\lambda$ in (11) is selected to be large, this formulation is close to the ERM problem, i.e., minimization of $\mathbb{E}_{\zeta \sim \hat{\mathbb{P}}}[l(\theta, \zeta)]$. Clearly, a distribution $\hat{\mathbb{P}}$ containing outliers would significantly deteriorate the learning performance of minimizing $\mathbb{E}_{\zeta \sim \hat{\mathbb{P}}}[l(\theta, \zeta)]$.

In contrast, when $\lambda$ is large, the problem (10) is close to the minimization of $\mathbb{E}_{\zeta \sim \hat{\mathbb{P}}}[\exp((l(\theta, \zeta) - h(\zeta)) / (\lambda \beta))] = \mathbb{E}_{\zeta \sim \hat{\mathbb{P}}}[\exp(l(\theta, \zeta) / (\lambda \beta)) w(\zeta)]$, where $w(\zeta) = \exp(-h(\zeta) / (\lambda \beta))$ acts as a re-weighting function. Recall that $h(\xi)$ is a function designed based on prior knowledge and assumes high values at outlier points. At an outlier point, say $\tilde{\xi}$, $h(\tilde{\xi})$ would be significantly larger than at non-outlier points, resulting in a very small weight $w(\tilde{\xi})$. Consequently, outlier data with small weights exert less influence on the resulting model. Therefore, the formulation (10) is more robust to outliers compared to standard WDRO.

**Comparison with outlier-robust WDRO [20].** [20] considers the Huber contamination model in which an $\varepsilon$ fraction of data can be arbitrarily corrupted. In [20], prior knowledge on the mean of the clean distribution is needed. Specifically, the function $h(\xi)$ that can identify outliers in [20] is selected as $h(\xi) = \lambda_1 \| \xi - \xi_0 \|^2$, where $\xi_0$ represents the estimated mean of the clean distribution and $\lambda_1 > 0$ is a tuning parameter. It can be verified that the corresponding Lagrangian formulation is

$$\min_{\theta \in \Theta} \mathrm{CVaR}_{1-\varepsilon, \hat{\mathbb{P}}} \left[ \sup_{\xi \in \Xi} \left\{ l(\theta, \xi) - \lambda_1 \| \xi - \xi_0 \|^2 - \lambda c(\xi, \zeta) \right\} \right], \tag{12}$$

where $\mathrm{CVaR}_{1-\varepsilon}$ denotes the conditional value at risk (CVaR), interpreted as the expected value of the rightmost $(1 - \varepsilon)$ percentile of outcomes[1].

Similarly, when $\lambda$ is very large, the formulation in (12) closely approximates the minimization of $\mathrm{CVaR}_{1-\varepsilon, \hat{\mathbb{P}}}[l(\theta, \zeta) - \lambda_1 \| \zeta - \xi_0 \|]$. At an outlier point, say $\tilde{\xi}$, the value $l(\theta, \tilde{\xi}) - \lambda_1 \| \tilde{\xi} - \xi_0 \|$ is small due to the large distance $\| \tilde{\xi} - \xi_0 \|$. Consequently, such an outlier point generates a function value that falls on the left tail of the loss distribution and is thus effectively excluded by the CVaR operation. This method can also be viewed as a re-weighting method by assigning weight 0 to outlier points and weighting other points equally. However, this approach is highly sensitive to the choice of $\varepsilon$, which is usually difficult to obtain a priori. In comparison, our method is a smoothed version of this CVaR method and is less sensitive to the prior knowledge of $\varepsilon$. Besides, calculating the value of CVaR is computationally more demanding when the number of samples is large as we will demonstrate in Sections 4 and 5.

# 4 Algorithm design

In this section, we focus on solving the dual of the Lagrangian penalty problem. We assume that the empirical distribution $\hat{\mathbb{P}}$ is constructed from $n$ samples and written as $\hat{\mathbb{P}} = \frac{1}{n} \sum_{i=1}^{n} \delta_{\hat{\zeta}_i}$ with $\hat{\zeta}_i$ the $i$-th sample. According to Theorem 2, the Lagrangian penalty problem is equivalent to the minimization of the following objective function

$$F(\theta) = \sum_{i=1}^{n} \exp \left( \sup_{\xi \in \Xi} \left\{ L(\theta, \xi) - \lambda c(\xi, \hat{\zeta}_i) \right\} / (\lambda \beta) \right). \tag{13}$$

---

[1] The CVaR of a random vector $Z$ that has the distribution $\mu$ with risk level $\varepsilon \in (0, 1)$ is defined as $\mathrm{CVaR}_{1-\varepsilon, \mu}[Z] = \inf_{\alpha} \alpha + \frac{1}{1-\varepsilon} \mathbb{E}_{Z \sim \mu}[Z - \alpha]_+$.

---

**Algorithm 1:** Distributionally robust optimization with outliers

---

**Require:** Sampling distribution $\hat{\mathbb{P}}$, constraint sets $\Theta$ and $Z$, step size sequence $\{\alpha_t > 0\}_{t=0}^{T-1}$

1: **for** $t = 1, \ldots, T$ **do**
2:    Sample $\hat{\zeta}_t$ uniformly from $\hat{\mathbb{P}}$
3:    Find an $\epsilon$-approximate maximizer $z_t$ of $\arg\max_{\xi \in \Xi} L(\theta_t, \xi) - \lambda c(\xi, \hat{\zeta}_t)$
4:    Construct gradient estimate $g_t = \exp\left(\left(L(\theta_t, z_t) - \lambda c(z_t, \hat{\zeta}_t)\right) / (\lambda\beta)\right) \frac{1}{\lambda\beta} \partial_\theta L(\theta_t, z_t)$
5:    $\theta_{t+1} = \text{Proj}_\Theta(\theta_t - \eta g_t)$
6: **end for**

---

To solve the problem (13), we employ the stochastic sub-gradient method, detailed in Algorithm 1. Specifically, at each time step $t$, we sample $\hat{\zeta}_t$ from the empirical distribution $\hat{\mathbb{P}}$ uniformly. We then find the $\epsilon$-approximate maximizer $z_t$ of the function $L(\theta_t, \xi) - \lambda c(\xi, \hat{\zeta}_t)$. We require that the solution satisfies $\text{dist}(z_t, X_t^*) \leq \epsilon$, where $X_t^*$ denotes the set of maximizers. This is solvable in many applications and can be achieved through, e.g., the (sub)-gradient method [27]. Then, we perform the projected sub-gradient update. The convergence analysis of Algorithm 1 is presented in the following theorem. The proof can be found in Appendix 7.4.

**Theorem 3.** *Suppose that $\Theta$ is a convex and compact set with bounded diameter $R$. Assume that the loss $L(\theta, \xi)$ is convex in $\theta$ for each $\xi \in \Xi$. For all data points $\zeta_i$ from the empirical distribution $\hat{\mathbb{P}}$, assume that $L(\theta, \cdot) - \lambda c(\cdot, \zeta_i)$ is concave, $L(\theta, \xi) - \lambda c(\xi, \zeta_i) \leq B$ and $|c(\xi, \zeta_i) - c(\xi', \zeta_i)| \leq L_c \|\xi - \xi'\|$, for all $\theta \in \Theta$, $\xi \in \Xi$, $\xi' \in \Xi$. Assume that $\|\partial_\theta L(\theta, \xi) - \partial_\theta L(\theta, \xi')\| \leq L_{\theta\xi} \|\xi - \xi'\|$, $\|\partial_\theta L(\theta, \xi)\| \leq B_2$, $\|L(\theta, \xi) - L(\theta, \xi')\| \leq L_\xi \|\xi - \xi'\|$. Select $\eta = \frac{1}{\sqrt{T}}$. Then, Algorithm 1 satisfies*

$$\frac{1}{T} \sum_{t=1}^{T} (\mathbb{E}[F(\theta_t)] - F^*) = O(\frac{1}{\sqrt{T}} + \epsilon), \tag{14}$$

*where $F^*$ represents the minimum value of the objective function* (13).

The accuracy parameter $\epsilon$ in the inner maximization problem has a fixed effect on the final optimization accuracy, independent of $T$. The assumptions in Theorem 3 are common in the analysis of optimization problems involving the exponential function in the objective function, see, e.g., [19, 28, 29].

**Algorithmic comparison with outlier-robust WDRO [20].** Our proposed formulation can be solved by the stochastic sub-gradient method, which is computationally more efficient when the sample size is large. In contrast, the approach in [20] utilizes CVaR to account for outliers and employs off-the-shelf solvers, e.g., GUROBI [33], to obtain a solution. A notable limitation of the presence of CVaR in DRO problems is that the number of constraints scales linearly with the sample size. Consequently, for large datasets, this scaling significantly increases the number of constraints, leading to computational inefficiency, as shown in Section 5. Designing a stochastic gradient method for the CVaR-based formulation in [20] is possible, although challenging since the estimate of the CVaR gradient is usually biased. Moreover, our methodology does not rely on off-the-shelf commercial solvers and can be incorporated in machine learning packages seamlessly, which allows us to handle a wide range of loss functions via the stochastic sub-gradient method.

## 5 Experiments

We conduct experiments on linear regression, linear classification, and logistic regression problems. We use the stochastic sub-gradient method in Algorithm 1 to solve the Lagrangian penalty problem and the GUROBI [33] solver to solve all other DRO problems we use as benchmarks. All experiments were conducted on an Intel Core i7-1185G7 CPU (3.00GHz) using Python 3.8. Our method is referred to as UOT-DRO. A discussion on parameter selection is provided at the end of this section.

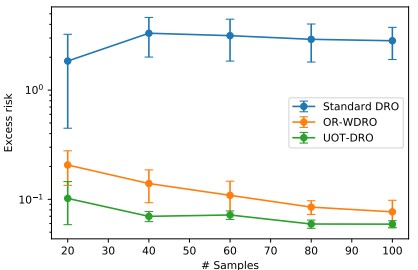
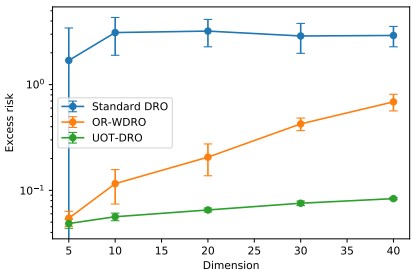

(a) Excess risk with various samples.    (b) Excess risk with varied dimensions.

Figure 1: Excess risk of standard DRO, OR-WDRO, and UOT-DRO with varied sample size and dimension for linear regression. The error bar denotes $\pm$ standard deviation.

## 5.1   Linear regression

We consider a linear regression problem with the loss $l_\theta(x, y) = |\theta^\mathsf{T} x - y|$, where $(x, y) \in \mathbb{R}^d \times \mathbb{R}$ represents a generic data point and $\theta \in \mathbb{R}^d$ is the model we aim to train. We define $\bar{\theta} = [\theta^\mathsf{T}, -1]^\mathsf{T}$, $\xi = [x^\mathsf{T}, y]^\mathsf{T}$. Then, the loss function can be written as $l_\theta(\xi) = |\bar{\theta}^\mathsf{T} \xi|$.

We generate a clean data distribution $\mathbb{P}_n$ with $n$ samples, which is uniform over $\{X_i, \theta_*^\mathsf{T} X_i\}_{i=1}^n$, where $X_1, \ldots, X_n$ are i.i.d. from $X \sim \mathcal{N}(0, I_d)$. Drawing a uniform random subset $S \subset [n]$ of size $\lfloor \varepsilon n \rfloor$, the corrupted data distribution $\hat{\mathbb{P}}_n$ is defined to be uniform over $\left\{ \left( C^{1\{i \in S\}} X_i, \left(-C^2\right)^{1\{i \in S\}} (\theta_*^\mathsf{T} X_i + \rho) \right) \right\}_{i=1}^n$, where $C > 0$ is a corruption scaling coefficient and $\rho > 0$ is a shift coefficient. The empirical probability distribution $\hat{\mathbb{P}}_n$ can be written as $\hat{\mathbb{P}}_n = \frac{1}{n} \sum_{i=1}^n \delta_{\hat{\zeta}_i}$ with $\hat{\zeta}_i$ the $i$-th sample.

We consider prior knowledge on the mean of the clean distribution, denoted as $\bar{\xi}$, and design the function $h(\xi) = \lambda_2 \left\| \xi - \bar{\xi} \right\|$. The value of $\bar{\xi}$ is determined by a cheap preprocessing step, same as in [20]. We consider the Lagrangian penalty problem which, based on Theorem 2, is given by

$$\min_\theta \mathbb{E}_{\zeta \sim \hat{\mathbb{P}}_n} \left[ \exp \left( \sup_\xi \left\{ |\bar{\theta}^\mathsf{T} \zeta| - \lambda \left\| \xi - \zeta \right\| - \lambda_2 \left\| \xi - \bar{\xi} \right\| \right\} / (\lambda \beta) \right) \right]. \tag{15}$$

Define $\kappa(\theta) = \sup\{ \|z\|_* : l^*(z) < \infty \} = \left\| \bar{\theta} \right\|_*$, where $\|\cdot\|_*$ denotes the dual norm. When $\lambda \geq \lambda_2 + \kappa(\theta)$, we can show that the problem (15) is equivalent to

$$\min_\theta \sum_{i=1}^N \exp \left( (|\bar{\theta}^\mathsf{T} \hat{\zeta}_i| - \lambda_2 ||\bar{\xi} - \hat{\zeta}_i||) / (\lambda \beta) \right). \tag{16}$$

The detailed proof is provided in Appendix 7.5.

We set the parameters as follows: $\lambda = 10$, $\beta = 6$, $\lambda_2 = 5$. All the results are averaged over 10 independent runs. We fix $\varepsilon = 0.1$ and $C = 8$. We compare the performance of our unbalanced DRO model against the standard DRO and outlier-robust WDRO provided in [20]. We evaluate these methods in terms of the excess risk, which we define as the difference between the loss incurred by the learned model and the minimum achievable loss. The simulation results presented in Fig. 1 demonstrate the performance of our method under varying conditions: In Fig. 1 (a), we set $d = 10$ and compare these methods for various samples sizes $n \in \{20, 40, 60, 80, 100\}$; In Fig. 1 (b), we fix $n = 100$ and explore the impact of various dimensions $d \in \{5, 10, 20, 30, 40\}$. We observe that our method not only achieves superior robustness to outliers but is also less sensitive to data dimensions.

Besides, we compare the computational efficiency of these methods across varying sample sizes. As shown in Table 1, the OR-WDRO method is not applicable to large-scale datasets, with its running time exceeding 200 minutes for a sample size of $n = 20000$. In contrast, our method only requires 20 seconds to process the same dataset while maintaining superior learning performance. Therefore, our method not only achieves better robustness but also maintains computational efficiency.

Table 1: Comparison of running time and excess risk of different methods for linear regression. The symbol '*' indicates that running time is over 12000 seconds.

| Sample Size $n$ | Standard DRO | | OR-WDRO [20] | | UOT-DRO | |
|---|---|---|---|---|---|---|
| | Time | Excess risk | Time | Excess risk | Time | Excess risk |
| 80 | 0.1 | 3.230 | 0.4 | 0.103 | 2.7 | 0.060 |
| 200 | 0.2 | 2.298 | 1.3 | 0.064 | 3.4 | 0.040 |
| 2000 | 3.3 | 0.441 | 29.8 | 0.050 | 4.7 | 0.038 |
| 5000 | 9.2 | 0.371 | 259.5 | 0.040 | 7.7 | 0.034 |
| 10000 | 28.9 | 0.352 | 1438.7 | 0.033 | 11.9 | 0.033 |
| 20000 | 110.8 | 0.380 | * | * | 22.2 | 0.031 |

Table 2: Excess risk with various contamination for linear classification.

| Contamin. $C$ | Standard DRO | | OR-WDRO [20] | | UOT-DRO | |
|---|---|---|---|---|---|---|
| | Excess risk | Accuracy | Excess risk | Accuracy | Excess risk | Accuracy |
| 6 | 0.628 | 73% | 0.627 | 72% | 0.298 | 92% |
| 10 | 0.722 | 66% | 0.560 | 79% | 0.295 | 93% |
| 30 | 0.637 | 68% | 0.341 | 90% | 0.241 | 96% |
| 100 | 0.872 | 56% | 0.191 | 97% | 0.240 | 95% |

## 5.2 Linear classification

We consider a linear classification problem with the loss $l_\theta(x, y) = \max\{0, 1 - y(\theta^\intercal x)\}$ with $\theta \in \mathbb{R}^d$. We consider the same outliers as that in [20]. Specifically, we first generate a clean distribution $\mathbb{P}_n$ by $\{X_i, \text{sign}(\theta_*^\intercal X_i)\}$, where $X_1, \ldots, X_n$ are drawn i.i.d. from $\mathcal{N}(0, I_d)$. For a uniform random subset $S \subset [n]$ of size $\lfloor \varepsilon n \rfloor$, we consider the corrupted distribution $\hat{\mathbb{P}}_n$ which is uniform over $\left\{ \left( (-C)^{1\{i \in S\}} X_i + e_1, \text{sign}(\theta_*^\intercal X_i) \right) \right\}_{i=1}^n$, where $C > 0$ is the contamination factor, $e_1$ is the distribution shift. We select $h(\xi) = \lambda_2 \left\| \xi - \bar{\xi} \right\|^2$ and $c(\xi, \zeta) = \left\| \xi - \zeta \right\|^2$ in linear classification as in [20].

We consider the formulation described in Theorem 2 and solve it using Algorithm 1. We fix $\varepsilon = 0.1$, $d = 10$, $n = 100$ and select $\lambda = 10$, $\lambda_2 = 1$, $\beta = 2$. The experimental results are averaged over 10 trials. We evaluate the model performance for different choices of the contamination factor $C$ by analyzing the excess risk and accuracy. Excess risk is defined as the difference between the returned loss and the best achievable loss. Accuracy refers to the rate of successful classification. As shown in Table 2, the performance of standard DRO decreases when $C$ gets large. In contrast, the OR-WDRO performs well only when $C$ is very large. This is because OR-WDRO uses CVaR to filter out outliers, which requires the outliers to be significantly distant from the regular data for CVaR to be effective. Nevertheless, our method performs well across the entire range of $C$.

## 5.3 Logistic regression

We consider a logistic regression with the loss function $l_\theta(x, y) = \log(1 + \exp(-y(\theta^\intercal X)))$, where $\theta \in \mathbb{R}^{10}$. As in [17], we assume that the feature vector $X$ follows a multivariate standard normal distribution, and the conditional distribution of the label $y \in \{-1, 1\}$ is given by $\text{Prob}(y|x) = [1 + \exp(-y(\theta_*^\intercal x))]^{-1}$, where $\theta_* = (10, 0, \ldots, 0)$. This setup uniquely determines the true distribution $\mathbb{P}$. We draw $n$ samples from this distribution to create the empirical distribution $\mathbb{P}_n = \frac{1}{n} \sum_{i=1}^n \delta_{(X_i, y_i)}$. Outliers are considered to occur only in the feature space. The perturbed distribution is represented as $\{(-C)^{1\{i \in S\}} X_i + \rho e_1\}_{i=1}^n$, where $S \subset [n]$ is a uniform random subset of size $\lfloor \varepsilon n \rfloor$. We fix $\varepsilon = 0.1$, $\rho = 0.1$, $e_1 = (1, 0, \ldots, 0)$, and select $\lambda = 10$, $\lambda_2 = 1$, $\beta = 2$. In Fig. 2, we compare the excess risk and accuracy of standard DRO as outlined in [17] and our proposed method, described by Theorem 2 and implemented by Algorithm 1. The results are averaged over 10 runs for different sample sizes $n \in \{20, 40, 60, 80, 100\}$. Besides, we conduct a comparative analysis of the excess risk and accuracy between the standard DRO and the proposed method across various contamination levels. As shown in Table 3, standard DRO is susceptible to significant outliers. Instead, the proposed method demonstrates robustness throughout the entire range of contamination factor $C$, consistently maintaining an accuracy of approximately 92%.

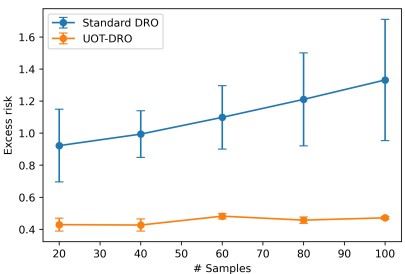
(a) Excess risk with various samples.

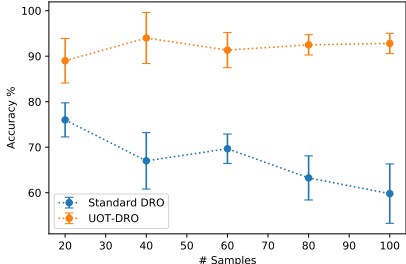
(b) Accuracy with various samples.

Figure 2: Excess risk and accuracy of standard DRO and UOT-DRO with varied sample sizes for logistic regression. The error bar denotes $\pm$ standard deviation.

Table 3: Loss and accuracy with various contamination for logistic regression.

| Contamination $C$ | Standard DRO [17] | | UOT-DRO | |
| --- | --- | --- | --- | --- |
| | Loss | Accuracy | Loss | Accuracy |
| 4 | 0.976 | 73% | 0.390 | 93% |
| 8 | 1.249 | 64% | 0.469 | 93% |
| 16 | 1.502 | 56% | 0.481 | 92% |
| 30 | 1.056 | 62% | 0.474 | 92% |

## 5.4 Parameter selection

We provide some guidelines for selecting the parameters $\lambda, \lambda_2, \beta$ in Algorithm 1. The parameter $\lambda$ is commonly used in the DRO literature [27] as a penalty coefficient. If the value of $\lambda$ is large, then the DRO problem approaches the empirical risk minimization problem, resulting in a model that performs well on the empirical distribution but is less robust to Wasserstein perturbations.

The parameter $\lambda_2$ represents the credibility level assigned to the function $h$. A larger value of $\lambda_2$ should be selected when the confidence in the reliability of $h$ is high, meaning that $h(\xi)$ is highly likely to become large at outlier points. Conversely, if the prior knowledge provided by $h$ is considered unreliable, the value of $\lambda_2$ should be reduced. If there is no reliable prior knowledge at all, in which case we should select $\lambda_2 = 0$, achieving good performance is impossible, as shown in related literature in robust statistics [18].

The parameter $\beta$ penalizes the mismatch between marginal distributions. A larger value of $\beta$ places more emphasis on minimizing this mismatch, possibly at the expense of increasing the transportation cost, thereby making the unbalanced optimal transport distance close to the balanced one. Conversely, a small value of $\beta$ allows for larger mismatches between the marginal distributions, which can enhance the model's robustness to outliers. However, when $\beta$ is very small, possible distribution mismatches incur little penalty, and the computed distance may fail to accurately represent the true distance between the distributions. In this case, the resulting DRO problem will incur many unlikely distributions in the ambiguity set, leading to a very conservative model.

## 6 Conclusion

In this work, we introduced a novel DRO framework that employs a new distance derived from UOT. By incorporating a soft penalization term in the design of the ambiguity set, our method exhibits increased resiliency to outliers. We provided strong duality results for the original DRO problem and the Lagrangian penalty problem, with the latter allowing for more efficient computation via stochastic sub-gradient methods. Finally, empirical results validate our method's enhanced robustness to outliers and reduced computational demands for regression and classification tasks.

## Acknowledgement

This work was supported in part by Swedish Research Council Distinguished Professor Grant 2017-01078, Knut and Alice Wallenberg Foundation, Wallenberg Scholar Grant, the Swedish Strategic Research Foundation SUCCESS Grant, and AFOSR under award #FA9550-19-1-0169.

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

# 7 Appendix

## 7.1 Proof of Lemma 1

By plugging in the definition of the UW distance in (4), we have that

$$(P) = \sup_{\mathbb{P},\bar{\mathbb{P}}\in\mathcal{M}(\Xi),\gamma\in\Gamma(\mathbb{P},\bar{\mathbb{P}})} \left\{ \mathbb{E}_{\xi\sim\mathbb{P}}[L(\theta,\xi)] : \mathbb{E}_{(\xi,\zeta)\sim\gamma}[c(\xi,\zeta)] + \beta D_{KL}(\bar{\mathbb{P}}||\hat{\mathbb{P}}) \leq \rho \right\}$$

$$= \sup_{\bar{\mathbb{P}},\gamma_\zeta\in\mathcal{M}(\Xi)} \left\{ \mathbb{E}_{\zeta\sim\bar{\mathbb{P}}}\mathbb{E}_{\xi\sim\gamma_\zeta}[L(\theta,\xi)] : \mathbb{E}_{\zeta\sim\bar{\mathbb{P}}}\mathbb{E}_{\xi\sim\gamma_\zeta}[c(\xi,\zeta)] + \mathbb{E}_{\zeta\sim\bar{\mathbb{P}}}\left[\log\frac{d\bar{\mathbb{P}}(\zeta)}{d\hat{\mathbb{P}}(\zeta)}\right] \leq \rho \right\}$$

$$= \sup_{\bar{\mathbb{P}},\gamma_\zeta\in\mathcal{M}(\Xi)} \left\{ \mathbb{E}_{\zeta\sim\bar{\mathbb{P}}}\mathbb{E}_{\xi\sim\gamma_\zeta}[L(\theta,\xi)] : \mathbb{E}_{\zeta\sim\bar{\mathbb{P}}}\mathbb{E}_{\xi\sim\gamma_\zeta}\left[c(\xi,\zeta) + \log\frac{d\bar{\mathbb{P}}(\zeta)}{d\hat{\mathbb{P}}(\zeta)}\right] \leq \rho \right\}$$

$$= \sup_{\bar{\mathbb{P}},\gamma_\zeta\in\mathcal{M}(\Xi)} \inf_{\lambda\geq 0} \mathbb{E}_{\zeta\sim\bar{\mathbb{P}}}\mathbb{E}_{\xi\sim\gamma_\zeta}[L(\theta,\xi)] + \lambda\left( \rho - \mathbb{E}_{\zeta\sim\bar{\mathbb{P}}}\mathbb{E}_{\xi\sim\gamma_\zeta}\left[c(\xi,\zeta) + \log\frac{d\bar{\mathbb{P}}(\zeta)}{d\hat{\mathbb{P}}(\zeta)}\right] \right)$$

$$\leq \inf_{\lambda\geq 0}\lambda\rho + \sup_{\bar{\mathbb{P}},\gamma_\zeta\in\mathcal{M}(\Xi)} \left\{ \mathbb{E}_{\zeta\sim\bar{\mathbb{P}}}\mathbb{E}_{\xi\sim\gamma_\zeta}\left[ L(\theta,\xi) - \lambda c(\xi,\zeta) - \lambda\beta\log\frac{d\bar{\mathbb{P}}(\zeta)}{d\hat{\mathbb{P}}(\zeta)} \right] \right\}$$

$$\leq \inf_{\lambda\geq 0}\lambda\rho + \sup_{\bar{\mathbb{P}}\in\mathcal{M}(\Xi)} \mathbb{E}_{\zeta\sim\bar{\mathbb{P}}}\left[ \sup_{\xi\in\Xi}\{L(\theta,\xi) - \lambda c(\xi,\zeta)\} - \lambda\beta\log\frac{d\bar{\mathbb{P}}(\zeta)}{d\hat{\mathbb{P}}(\zeta)} \right] \tag{17}$$

where the second last inequality follows from the max-min inequality and the last inequality follows since $\mathbb{E}_{\xi\sim\gamma_\zeta}[L(\theta,\xi) - \lambda c(\xi,\zeta)] \leq \sup_{\xi\in\Xi} L(\theta,\xi) - \lambda c(\xi,\zeta)$. Recalling the definition $f_\lambda(\theta,\zeta) = \sup_{\xi\in\Xi}\{L(\theta,\xi) - \lambda c(\xi,\zeta)\}$, we obtain that

$$\inf_{\lambda\geq 0}\lambda\rho + \sup_{\bar{\mathbb{P}}\in\mathcal{M}(\Xi)} \mathbb{E}_{\zeta\sim\bar{\mathbb{P}}}\left[ f_\lambda(\theta,\zeta) - \lambda\beta\log\frac{d\bar{\mathbb{P}}(\zeta)}{d\hat{\mathbb{P}}(\zeta)} \right] = \inf_{\lambda\geq 0}\lambda\rho + \lambda\beta\log\mathbb{E}_{\zeta\sim\hat{\mathbb{P}}}\left[ \exp\left(\frac{f_\lambda(\theta,\zeta)}{\lambda\beta}\right) \right],$$

where the last equality follows from Donsker and Varadhan's variational formula [34] and the fact that $f_\lambda(\theta,\zeta)$ is integrable with respect to $\hat{\mathbb{P}}$. Note that the condition $|f_\lambda(\theta,\zeta)| < \infty$ for $\hat{\mathbb{P}}$-almost every $\zeta$ ensures $\frac{\exp(f_\lambda(\theta,\zeta)/(\lambda\beta))}{\mathbb{E}_{\zeta\sim\hat{\mathbb{P}}}[\exp(f_\lambda(\theta,\zeta)/(\lambda\beta))]}d\hat{\mathbb{P}}(\zeta)$ is well-defined. The supremum is achieved by $d\bar{\mathbb{P}}(\zeta) = \frac{\exp(f_\lambda(\theta,\zeta)/(\lambda\beta))}{\mathbb{E}_{\zeta\sim\hat{\mathbb{P}}}[\exp(f_\lambda(\theta,\zeta)/(\lambda\beta))]}d\hat{\mathbb{P}}(\zeta)$. The proof is complete.

## 7.2 Proof of Theorem 1

Given Assumption 2–3, $L(\theta,\xi) - \lambda c(\xi,\zeta)$ is differentiable with respect to $\xi$. According to the Envelope theorem [35], we have $f_\lambda(\zeta)$ is differentiable in $\lambda$. Given $L(\theta,\xi) - \lambda c(\xi,\zeta)$ is strongly concave in $\xi$, we have $\nabla_\lambda f_\lambda(\zeta) = -c(h_\lambda(\zeta),\zeta)$, where $h_\lambda(\zeta) := \arg\max_{\xi\in\Xi}\{L(\theta,\xi) - \lambda c(\xi,\zeta)\}$.

Define $v(\lambda) = \lambda\rho + \lambda\beta\log\mathbb{E}_{\zeta\sim\hat{\mathbb{P}}}\left[ \exp\left(\frac{f_\lambda(\zeta)}{\lambda\beta}\right) \right]$. We have $v(\lambda)$ is differentiable and satisfies

$$\nabla_\lambda v(\lambda) = \rho + \beta\log\mathbb{E}_{\zeta\sim\hat{\mathbb{P}}}\left[ \exp\left(\frac{f_\lambda(\zeta)}{\lambda\beta}\right) \right] + \frac{\mathbb{E}_{\zeta\sim\hat{\mathbb{P}}}\left[ \exp\left(\frac{f_\lambda(\zeta)}{\lambda\beta}\right)\frac{\nabla_\lambda f_\lambda(\zeta)\lambda - f_\lambda(\zeta)}{\lambda} \right]}{\mathbb{E}_{\zeta\sim\hat{\mathbb{P}}}\left[ \exp\left(\frac{f_\lambda(\zeta)}{\lambda\beta}\right) \right]}. \tag{18}$$

Since $\lambda^* > 0$, due to first-order optimality condition, we must have $\nabla_\lambda(v(\lambda))|_{\lambda=\lambda^*} = 0$, i.e.,

$$\rho + \beta\log\mathbb{E}_{\zeta\sim\hat{\mathbb{P}}}\left[ \exp\left(\frac{f_{\lambda^*}(\zeta)}{\lambda^*\beta}\right) \right] + \frac{\mathbb{E}_{\zeta\sim\hat{\mathbb{P}}}\left[ \exp\left(\frac{f_{\lambda^*}(\zeta)}{\lambda^*\beta}\right)\left(-l(h_{\lambda^*}(\zeta))\right) \right]}{\lambda^*\mathbb{E}_{\zeta\sim\hat{\mathbb{P}}}\left[ \exp\left(\frac{f_{\lambda^*}(\zeta)}{\lambda^*\beta}\right) \right]} = 0. \tag{19}$$

We define the joint distribution $\gamma^*$ such that $d\gamma^*(\xi,\zeta) = d\gamma_\zeta^*(\xi)d\bar{\mathbb{P}}^*(\zeta)$, where $d\bar{\mathbb{P}}^*(\zeta) = \frac{\exp(f_{\lambda^*}(\zeta)/(\lambda^*\beta))}{\mathbb{E}_{\zeta\sim\hat{\mathbb{P}}}[\exp(f_{\lambda^*}(\zeta)/(\lambda^*\beta))]}d\hat{\mathbb{P}}(\zeta)$, and $\gamma_\zeta^*(\xi^*) = 1$, if $\xi^* = h_{\lambda^*}(\zeta) = \arg\max_\xi\{L(\theta,\xi) - \lambda^* c(\xi,\zeta)\}$

and $\gamma_\zeta^*(\xi) = 0$ otherwise. Let $\mathbb{P}^*$ be the first marginal distribution of $\gamma^*$, i.e., it satisfies that $\gamma^* \in \Gamma(\mathbb{P}^*, \bar{\mathbb{P}}^*)$. In what follows, we show that $\mathbb{P}^*$ lies in the ambiguity set. Specifically, we have

$$
\begin{aligned}
\mathrm{UW}(\mathbb{P}^*||\hat{\mathbb{P}}) &= \inf_{\bar{\mathbb{P}}, \gamma \in \Gamma(\mathbb{P}^*, \bar{\mathbb{P}})} \left\{ \mathbb{E}_{(\xi,\zeta)\sim\gamma}[c(\xi,\zeta)] + \beta D_{KL}(\bar{\mathbb{P}}||\hat{\mathbb{P}}) \right\} \\
&= \inf_{\bar{\mathbb{P}}, \gamma \in \Gamma(\mathbb{P}^*, \bar{\mathbb{P}})} \left\{ \mathbb{E}_{(\xi,\zeta)\sim\gamma} \left[ c(\xi,\zeta) + \beta \log \frac{\mathrm{d}\bar{\mathbb{P}}(\zeta)}{\mathrm{d}\hat{\mathbb{P}}(\zeta)} \right] \right\} \\
&\leq \mathbb{E}_{(\xi,\zeta)\sim\gamma^*} \left[ c(\xi,\zeta) + \beta \log \frac{\mathrm{d}\bar{\mathbb{P}}^*(\zeta)}{\mathrm{d}\hat{\mathbb{P}}(\zeta)} \right] \\
&= \mathbb{E}_{\zeta\sim\bar{\mathbb{P}}^*} \mathbb{E}_{\xi\sim\gamma_\zeta^*}[c(\xi,\zeta)] + \beta \mathbb{E}_{\zeta\sim\bar{\mathbb{P}}^*} \left[ \log \frac{\mathrm{d}\bar{\mathbb{P}}^*(\zeta)}{\mathrm{d}\hat{\mathbb{P}}(\zeta)} \right] \\
&= \mathbb{E}_{\zeta\sim\bar{\mathbb{P}}^*}[c(h_{\lambda^*}(\zeta),\zeta)] + \beta \mathbb{E}_{\zeta\sim\bar{\mathbb{P}}^*} \left[ \log \frac{\exp\left(\frac{f_{\lambda^*}(\zeta)}{\lambda^*\beta}\right)}{\mathbb{E}_{\zeta\sim\hat{\mathbb{P}}}\left[\exp\left(\frac{f_{\lambda^*}(\zeta)}{\lambda^*\beta}\right)\right]} \right] \\
&= \mathbb{E}_{\zeta\sim\bar{\mathbb{P}}^*} \left[ c(h_{\lambda^*}(\zeta),\zeta) + \frac{f_{\lambda^*}(\zeta)}{\lambda^*} \right] - \beta \log \mathbb{E}_{\zeta\sim\bar{\mathbb{P}}^*} \left[ \exp\left(\frac{f_{\lambda^*}(\zeta)}{\lambda^*\beta}\right) \right] \\
&= \mathbb{E}_{\zeta\sim\bar{\mathbb{P}}^*} \left[ \frac{l(h_{\lambda^*}(\zeta))}{\lambda^*} \right] - \beta \log \mathbb{E}_{\zeta\sim\bar{\mathbb{P}}^*} \left[ \exp\left(\frac{f_{\lambda^*}(\zeta)}{\lambda^*\beta}\right) \right] \\
&= \mathbb{E}_{\zeta\sim\hat{\mathbb{P}}} \left[ \frac{l(h_{\lambda^*}(\zeta))}{\lambda^*} \frac{\mathrm{d}\bar{\mathbb{P}}^*(\zeta)}{\mathrm{d}\hat{\mathbb{P}}(\zeta)} \right] - \beta \log \mathbb{E}_{\zeta\sim\bar{\mathbb{P}}^*} \left[ \exp\left(\frac{f_{\lambda^*}(\zeta)}{\lambda^*\beta}\right) \right] \\
&= \frac{\mathbb{E}_{\zeta\sim\hat{\mathbb{P}}} \left[ \exp\left(\frac{f_{\lambda^*}(\zeta)}{\lambda^*\beta}\right) l(h_{\lambda^*}(\zeta)) \right]}{\lambda^* \mathbb{E}_{\zeta\sim\hat{\mathbb{P}}} \left[ \exp\left(\frac{f_{\lambda^*}(\zeta)}{\lambda^*\beta}\right) \right]} - \beta \log \mathbb{E}_{\zeta\sim\bar{\mathbb{P}}^*} \left[ \exp\left(\frac{f_{\lambda^*}(\zeta)}{\lambda^*\beta}\right) \right] \\
&= \rho,
\end{aligned}
\tag{20}
$$

where the inequality holds since $\gamma^*$ is a feasible solution in $\Gamma(\mathbb{P}^*, \bar{\mathbb{P}})$. The following equalities are derived by substituting the expressions of $\gamma^*$ and $\bar{\mathbb{P}}^*$. The last equality follows due to the first-order optimality condition (19). Therefore, the distribution $\mathbb{P}^*$ is feasible for the primal problem. Then, the primal optimal value is lower bounded by the value

$$
\begin{aligned}
(\mathrm{P}) &\geq \mathbb{E}_{\xi\sim\mathbb{P}^*}[L(\theta,\xi)] = \mathbb{E}_{(\xi,\zeta)\sim\gamma^*}[L(\theta,\xi)] \\
&= \mathbb{E}_{\zeta\sim\bar{\mathbb{P}}^*} \mathbb{E}_{\xi\sim\gamma_\zeta^*}[L(\theta,\xi)] \\
&= \mathbb{E}_{\zeta\sim\bar{\mathbb{P}}^*}[l(h_{\lambda^*}(\zeta))] \\
&= \mathbb{E}_{\zeta\sim\hat{\mathbb{P}}} \left[ l(h_{\lambda^*}(\zeta)) \frac{\mathrm{d}\bar{\mathbb{P}}^*(\zeta)}{\mathrm{d}\hat{\mathbb{P}}(\zeta)} \right] \\
&= \mathbb{E}_{\zeta\sim\hat{\mathbb{P}}} \left[ l(h_{\lambda^*}(\zeta)) \frac{\exp\left(\frac{f_{\lambda^*}(\zeta)}{\lambda^*\beta}\right)}{\mathbb{E}_{\zeta\sim\hat{\mathbb{P}}}\left[\exp\left(\frac{f_{\lambda^*}(\zeta)}{\lambda^*\beta}\right)\right]} \right] \\
&= \lambda^*\rho + \lambda^*\beta \mathbb{E}_{\zeta\sim\hat{\mathbb{P}}} \left[ \exp\left(\frac{f_{\lambda^*}(\zeta)}{\lambda^*\beta}\right) \right] = (\mathrm{D}).
\end{aligned}
\tag{21}
$$

Combining (21) with the weak duality, we have $(\mathrm{P}) = (\mathrm{D})$. Thus, the strong duality holds when $\lambda^* > 0$. The proof is complete.

## 7.3 Proof of Theorem 2

Based on the definition of UW in (4), we have

$$
\sup_{\mathbb{P}\in\mathcal{M}(\Xi)} \mathbb{E}_{\xi\sim\mathbb{P}}[L(\theta,\xi)] - \lambda\mathrm{UW}(\mathbb{P}||\hat{\mathbb{P}})
$$

$$
= \sup_{\mathbb{P}\in\mathcal{M}(\Xi)} \mathbb{E}_{\xi\sim\mathbb{P}}[L(\theta,\xi)] - \lambda \inf_{\bar{\mathbb{P}},\gamma\in\Gamma(\mathbb{P},\bar{\mathbb{P}})} \left\{ \mathbb{E}_{(\xi,\varsigma)\sim\gamma}[c(\xi,\varsigma)] + \beta D_{KL}(\bar{\mathbb{P}}||\hat{\mathbb{P}}) \right\}
$$

$$
= \sup_{\bar{\mathbb{P}}\in\mathcal{M}(\Xi),\gamma\in\Gamma(\mathbb{P},\bar{\mathbb{P}})} \mathbb{E}_{\xi\sim\mathbb{P}}[L(\theta,\xi)] - \lambda\mathbb{E}_{(\xi,\varsigma)\sim\gamma}[c(\xi,\varsigma)] - \lambda\beta\mathbb{E}_{\varsigma\sim\bar{\mathbb{P}}}\left[ \log\frac{d\bar{\mathbb{P}}(\varsigma)}{d\hat{\mathbb{P}}(\varsigma)} \right]
$$

$$
= \sup_{\bar{\mathbb{P}},\gamma_\varsigma\in\mathcal{M}(\Xi)} \mathbb{E}_{\varsigma\sim\bar{\mathbb{P}}}\mathbb{E}_{\xi\sim\gamma_\varsigma}\left[ L(\theta,\xi) - \lambda c(\xi,\varsigma) - \lambda\beta\log\frac{d\bar{\mathbb{P}}(\varsigma)}{d\hat{\mathbb{P}}(\varsigma)} \right]
$$

$$
= \sup_{\bar{\mathbb{P}}\in\mathcal{M}(\Xi)} \mathbb{E}_{\varsigma\sim\bar{\mathbb{P}}}\left[ \sup_{\xi}\{L(\theta,\xi) - \lambda c(\xi,\varsigma)\} - \lambda\beta\log\frac{d\bar{\mathbb{P}}(\varsigma)}{d\hat{\mathbb{P}}(\varsigma)} \right]. \tag{22}
$$

The last equality in (22) is achieved by selecting $\gamma_\varsigma^*(\xi^*) = 1$ for a $\xi^* \in \arg\max_{\xi\in\Xi}\{L(\theta,\xi) - \lambda c(\xi,\varsigma)\}$ and $\gamma_\varsigma^*(\xi) \in 0$ otherwise. Here, there always exists such a $\xi^*$ since $L(\theta,\xi) - \lambda c(\xi,\varsigma)$ is continuous and $\Xi$ is a compact set. Recalling the definition of $f_\lambda(\varsigma)$, we have

$$
\sup_{\mathbb{P}\in\mathcal{M}(\Xi)} \mathbb{E}_{\xi\sim\mathbb{P}}[L(\theta,\xi)] - \lambda\mathrm{UW}(\mathbb{P}||\hat{\mathbb{P}})
$$

$$
= \sup_{\bar{\mathbb{P}}\in\mathcal{M}(\Xi)} \mathbb{E}_{\varsigma\sim\bar{\mathbb{P}}}\left[ f_\lambda(\varsigma) - \lambda\beta\log\frac{d\bar{\mathbb{P}}(\varsigma)}{d\hat{\mathbb{P}}(\varsigma)} \right]
$$

$$
= \lambda\beta\log\mathbb{E}_{\varsigma\sim\hat{\mathbb{P}}}\left[ \exp\left(\frac{f_\lambda(\varsigma)}{\lambda\beta}\right) \right], \tag{23}
$$

where the last equality follows from [14]. The proof ends.

## 7.4 Proof of Theorem 3

Since $L(\theta,\xi) - \lambda c(\xi,\hat{\zeta}_t)$ is convex in $\theta$, and $\Xi$ is a compact set, from Danskin's theorem [36], we have $f_\lambda(\theta,\hat{\zeta}_t)$ is convex in $\theta$ and

$$
\partial_\theta f_\lambda(\theta_t,\hat{\zeta}_t) = \mathrm{Co}\cup\{\partial_\theta L(\theta_t,z^*)|f_\lambda(\theta_t,\hat{\zeta}_t) = L(\theta_t,z^*) - \lambda c(z^*,\hat{\zeta}_t)\}, \tag{24}
$$

where $\mathrm{Co}$ denotes the convex hull of a point set. Given that $f_\lambda(\theta,\hat{\zeta}_t)$ is convex, it is easy to verify that $F(\theta)$ is convex. Since the exponential function is convex and non-decreasing, for any $z_t^* \in \arg\max_\xi L(\theta_t,\xi) - \lambda c(\xi,\hat{\zeta}_t)$, we have

$$
G_t = \exp((L(\theta_t,z_t^*) - \lambda c(z_t^*,\hat{\zeta}_t))/(\lambda\beta))\frac{1}{\lambda\beta}\partial_\theta L(\theta_t,z_t^*) \subset \partial_\theta\exp(f_\lambda(\theta_t,\hat{\zeta}_t)/(\lambda\beta)).
$$

Note that $\|G_t\| \leq \exp(\frac{B}{\lambda\beta})\frac{B_2}{\lambda\beta} := C_0$. Since we obtain $\epsilon$-approximate maximizer $z_t$ of $\arg\max_\xi L(\theta_t,\xi) - \lambda c(\xi,\hat{\zeta}_t)$, the sub-gradient $g_t$ is inexact and we denote by the error $e_t = g_t - G_t$.

In what follows, we analyze the bound of sub-gradient error $e_t$. From its definition, we have

$$\|e_t\| = \left\| e^{(L(\theta_t, z_t) - \lambda c(z_t, \hat{\zeta}_t))/(\lambda\beta)} \frac{1}{\lambda\beta} \partial_\theta L(\theta_t, z_t) - e^{(L(\theta_t, z_t^*) - \lambda c(z_t^*, \hat{\zeta}_t))/(\lambda\beta)} \frac{1}{\lambda\beta} \partial_\theta L(\theta_t, z_t^*) \right\|$$

$$= \left\| \frac{1}{\lambda\beta} \Big( e^{(L(\theta_t, z_t) - \lambda c(z_t, \hat{\zeta}_t))/(\lambda\beta)} \big( \partial_\theta L(\theta_t, z_t) - \partial_\theta L(\theta_t, z_t^*) \big) \right.$$

$$\left. + \partial_\theta L(\theta_t, z_t^*) \big( e^{(L(\theta_t, z_t) - \lambda c(z_t, \hat{\zeta}_t))/(\lambda\beta)} - e^{(L(\theta_t, z_t^*) - \lambda c(z_t^*, \hat{\zeta}_t))/(\lambda\beta)} \big) \Big) \right\|$$

$$\leq \frac{1}{\lambda\beta} \left\| e^{(L(\theta_t, z_t) - \lambda c(z_t, \hat{\zeta}_t))/(\lambda\beta)} \big( \partial_\theta L(\theta_t, z_t) - \partial_\theta L(\theta_t, z_t^*) \big) \right\|$$

$$+ \frac{1}{\lambda\beta} \left\| \partial_\theta L(\theta_t, z_t^*) \big( e^{(L(\theta_t, z_t) - \lambda c(z_t, \hat{\zeta}_t))/(\lambda\beta)} - e^{(L(\theta_t, z_t^*) - \lambda c(z_t^*, \hat{\zeta}_t))/(\lambda\beta)} \big) \right\|$$

$$\leq \frac{e^{\frac{B}{\lambda\beta}} L_{\theta\xi} \|z_t - z_t^*\|}{\lambda\beta} + \frac{1}{\lambda\beta} \left\| \partial_\theta L(\theta_t, z_t^*) \big( e^{(L(\theta_t, z_t) - \lambda c(z_t, \hat{\zeta}_t))/(\lambda\beta)} - e^{(L(\theta_t, z_t^*) - \lambda c(z_t^*, \hat{\zeta}_t))/(\lambda\beta)} \big) \right\|$$

$$\leq \frac{e^{\frac{B}{\lambda\beta}} L_{\theta\xi}}{\lambda\beta} \|z_t - z_t^*\| + \frac{B_2}{\lambda\beta} e^B \left\| \frac{L(\theta_t, z_t) - \lambda c(z_t, \hat{\zeta}_t)}{\lambda\beta} - \frac{L(\theta_t, z_t^*) - \lambda c(z_t^*, \hat{\zeta}_t)}{\lambda\beta} \right\|$$

$$\leq \left( \frac{e^{\frac{B}{\lambda\beta}} L_{\theta\xi}}{\lambda\beta} + \frac{B_2 e^B (L_\xi + L_c)}{\lambda^2 \beta^2} \right) \|z_t - z_t^*\|$$

$$:= C_1 \|z_t - z_t^*\|, \tag{25}$$

where the first inequality follows from triangle inequality, the second inequality holds since $L(\theta, \xi) - \lambda c(\xi, \hat{\zeta}_t) \leq B$ and $\|\partial_\theta L(\theta_t, z_t) - \partial_\theta L(\theta_t, z_t^*)\| \leq L_{\theta\xi} \|z_t - z_t^*\|$. The third inequality hols since $|e^x - e^y| \leq e^B |x - y|$ when $x, y < B$. The last inequality follows from our assumptions. Since (25) holds for any $z_t^*$, we have $\|e_t\| \leq C_1\epsilon$.

Suppose that $\theta^*$ minimizes $F(\theta)$. Based on the update rule, we have

$$\|\theta_{t+1} - \theta^*\|^2 \leq \|\theta_t - \eta g_t - \theta^*\|^2$$
$$= \|\theta_t - \eta G_t - \eta e_t - \theta^*\|^2$$
$$= \|\theta_t - \theta^*\|^2 + \eta^2 \|G_t\|^2 + \eta^2 \|e_t\|^2 - 2\langle \theta_t - \theta^*, \eta G_t \rangle - 2\langle \theta_t - \theta^*, \eta e_t \rangle + \eta^2 \langle G_t, e_t \rangle. \tag{26}$$

Taking conditional expectation on both sides of (26) with respect to $\hat{\zeta}_t$, we have

$$\mathbb{E} \|\theta_{t+1} - \theta^*\|^2$$
$$\leq \|\theta_t - \theta^*\|^2 + \eta^2 \mathbb{E} \|G_t\|^2 + \eta^2 \mathbb{E} \|e_t\|^2 - 2\langle \theta_t - \theta^*, \eta \mathbb{E} G_t \rangle - 2\langle \theta_t - \theta^*, \eta \mathbb{E} e_t \rangle + \eta^2 \langle G_t, e_t \rangle$$
$$\leq \|\theta_t - \theta^*\|^2 - 2\eta(\mathbb{E} F(\theta_t) - F^*) + \eta^2 \mathbb{E} \|G_t\|^2 + \eta^2 \mathbb{E} \|e_t\|^2 - 2\langle \theta_t - \theta^*, \eta \mathbb{E} e_t \rangle + \eta^2 \langle G_t, e_t \rangle$$
$$\leq \|\theta_t - \theta^*\|^2 - 2\eta(\mathbb{E} F(\theta_t) - F^*) + \eta^2 C_0^2 + \eta^2 \mathbb{E} \|e_t\|^2 - 2\langle \theta_t - \theta^*, \eta \mathbb{E} e_t \rangle + 2\eta^2 C_0 \|e_t\|$$
$$\leq \|\theta_t - \theta^*\|^2 - 2\eta(\mathbb{E} F(\theta_t) - F^*) + \eta^2 C_0^2 + \eta^2 C_1^2 \epsilon^2 + 2\eta C_1 R\epsilon + 2\eta^2 C_0 C_1 \epsilon, \tag{27}$$

where the second inequality holds since $\mathbb{E}[G_t] \subset \partial F(\theta_t)$, the third inequality follows from the fact that $G_t \leq C_0$ and the last inequality follow since $\|e_t\| \leq C_1\epsilon$. Taking expectation on both sides of (27) and summing up over $t = 1, \ldots, T$, it gives

$$\mathbb{E} \|\theta_{T+1} - \theta^*\|^2$$
$$\leq \mathbb{E} \|\theta_1 - \theta^*\|^2 - 2\sum_{t=1}^T \eta(\mathbb{E} F(\theta_t) - F^*) + T(\eta^2 C_0^2 + \eta^2 C_1^2 \epsilon^2 + 2\eta C_1 R\epsilon + 2\eta^2 C_0 C_1 \epsilon). \tag{28}$$

Rearranging (28), we have

$$\frac{1}{T} \sum_{t=1}^T (\mathbb{E} F(\theta_t) - F^*) \leq \frac{R^2}{2\eta T} + \frac{\eta C_0^2}{2} + \frac{\eta C_1^2 \epsilon^2}{2} + C_1 R\epsilon + \eta C_0 C_1 \epsilon. \tag{29}$$

Substituting $\eta = \frac{1}{\sqrt{T}}$ into (29) completes the proof.

## 7.5 Additional details of linear regression experiment

In this section, we show the derivation of equation (16).

Recall that $\bar{\theta} = [\theta^{\mathsf{T}}, -1]^{\mathsf{T}}$, $\zeta = [x^{\mathsf{T}}, y]^{\mathsf{T}}$ and $l_\theta(\xi) = |\bar{\theta}^{\mathsf{T}}\xi|$. As $l_\theta$ is proper, convex, and lower semicontinuous, it coincides with its bi-conjugate function $l^{**}$, see e.g. [7]. Thus, we can write $l_\theta(\xi) = \sup_{z \in \mathcal{Z}} \langle z, \xi \rangle - l_\theta^*(z)$, where $\mathcal{Z} = \{z \in \mathbb{R}^d : l^*(z) < \infty$ is the effective domain of the conjugate function. Define $\kappa(\theta) = \sup\{\|z\|_* : z \in \mathcal{Z}\}$. It is shown by [16] that $\kappa(\theta) = \|\bar{\theta}\|_*$. With the definition of the dual norm, we have

$$\sup_{\xi \in \mathbb{R}^d} \left\{ l_\theta(\xi) - \lambda \|\xi - \zeta\| - \lambda_2 \|\xi - \bar{\xi}\| \right\}$$

$$= \sup_{\xi \in \mathbb{R}^d} \sup_{z \in \mathcal{Z}} \left\{ \langle z, \xi \rangle - l_\theta^*(z) - \lambda \|\xi - \zeta\| - \lambda_2 \|\xi - \bar{\xi}\| \right\}$$

$$= \sup_{\xi \in \mathbb{R}^d} \sup_{z \in \mathcal{Z}} \inf_{\|z_1\|_* \leq \lambda, \|z_2\|_* \leq \lambda_2} \left\{ \langle z, \xi \rangle - l_\theta^*(z) - \langle z_1, \xi - \hat{\zeta}_i \rangle - \langle z_2, \xi - \bar{\xi} \rangle \right\}.$$

The classical minimax theorem [37] allows us to interchange the maximization and minimization and then we obtain

$$\sup_{\xi \in \mathbb{R}^d} \left\{ l_\theta(\xi) - \lambda \|\xi - \zeta\| - \lambda_2 \|\xi - \bar{\xi}\| \right\}$$

$$= \sup_{z \in \mathcal{Z}} \inf_{\|z_1\|_* \leq \lambda, \|z_2\|_* \leq \lambda_2} \sup_{\xi \in \mathbb{R}^d} \left\{ \langle z + z_1 + z_2, \xi \rangle - l_\theta^*(z) - \langle z_1, \hat{\zeta}_i \rangle - \langle z_2, \bar{\xi} \rangle \right\}$$

$$= \sup_{z \in \mathcal{Z}} \inf_{\|z_1\|_* \leq \lambda, \|z_2\|_* \leq \lambda_2} \left\{ \mathcal{X}_{\{0\}}(z + z_1 + z_2) - l_\theta^*(z) - \langle z_1, \hat{\zeta}_i \rangle - \langle z_2, \bar{\xi} \rangle \right\}$$

$$= \inf_{\|z_1\|_* \leq \lambda, \|z_2\|_* \leq \lambda_2, z+z_1+z_2=0} \sup_{z \in \mathcal{Z}} \left\{ -l_\theta^*(z) - \langle z_1, \hat{\zeta}_i \rangle - \langle z_2, \bar{\xi} \rangle \right\}$$

$$= \inf_{\|z_1\|_* \leq \lambda, \|z_2\|_* \leq \lambda_2, z+z_1+z_2=0} \sup_{z \in \mathcal{Z}} \left\{ -l_\theta^*(z) + \langle z, \hat{\zeta}_i \rangle + \langle z_2, \hat{\zeta}_i - \bar{\xi} \rangle \right\}$$

$$= \inf_{\|z_1\|_* \leq \lambda, \|z_2\|_* \leq \lambda_2, z+z_1+z_2=0, z \in \mathcal{Z}} \left\{ l_\theta(\hat{\zeta}_i) + \langle z_2, \hat{\zeta}_i - \bar{\xi} \rangle \right\}$$

$$= \inf_{\|z_1\|_* \leq \lambda, \|z_2\|_* \leq \lambda_2, z+z_1+z_2=0, z \in \mathcal{Z}} \left\{ l_\theta(\hat{\zeta}_i) - \lambda_2 \left\| \hat{\zeta}_i - \bar{\xi} \right\|_* \right\}.$$

The second equality holds since $\sup_{\xi \in \mathbb{R}^d} \langle z, \xi \rangle = \sigma_{\mathbb{R}^d}(z) = \mathcal{X}_0$, where $\sigma_\Omega$ and $\mathcal{X}_\Omega$ denote the support function and characteristic function of the set $\Omega$, respectively. The third equality follows again from the classical minimax theorem [37]. The fourth equality is derived by substituting $z_1 = -z_2 - z$ into the objective function. The fifth equality holds since $l_\theta = l_\theta^{**}$. Since $\lambda > \lambda_2 + \kappa(\theta)$, there exists $z_2$ that satisfies $z_2 = -z_1 - z$ and other constraints. Finally, we obtain

$$\sup_{\xi \in \mathbb{R}^d} \left\{ l_\theta(\xi) - \lambda \|\xi - \zeta\| - \lambda_2 \|\xi - \bar{\xi}\| \right\} = l_\theta(\hat{\zeta}_i) - \lambda_2 \left\| \hat{\zeta}_i - \bar{\xi} \right\|_*.$$

The proof is complete.

## 7.6 Additional experiments on linear regression

We evaluate the sensitivity of the model performance to the tuning of designed parameters. We first fix $\lambda_2 = 5$ and explore the impact of varying $\lambda$ and $\beta$. As shown in Table 4, the model's performance demonstrates minimal sensitivity to changes in these parameters. Then, we fix $\beta = 6$, and assess the sensitivity to changes in $\lambda_2$. The simulation results are presented in Table 5. When $\lambda_2 = 0$, meaning we abandon the use of the prior knowledge term $\lambda_2 \left\| \bar{\xi} - \hat{\zeta}_i \right\|$, our method performs worse than the traditional DRO method. This is because the unbalanced optimal transport distance itself includes distributions that contain worse outliers, thereby making our model overly conservative. As $\lambda_2$ increases, there is a noticeable improvement in learning performance, which we attribute to the enhanced role of prior knowledge in outlier detection.

We also explore how the contamination factor, denoted as $C$, affects model performance. The simulation result is presented in Table 6. While standard DRO is highly sensitive to the contamination

Table 4: Excess risk with various parameters for linear regression.

| Excess Risk | $\lambda = 6$ | $\lambda = 8$ | $\lambda = 12$ |
|---|---|---|---|
| $\beta = 1$ | 0.064 | 0.059 | 0.060 |
| $\beta = 6$ | 0.057 | 0.056 | 0.058 |
| $\beta = 10$ | 0.059 | 0.055 | 0.060 |

Table 5: Excess risk with various $\lambda_2$ for linear regression.

| $\lambda_2$ | Standard DRO | UOT-DRO ($\lambda = 8$) | UOT-DRO ($\lambda = 4$) |
|---|---|---|---|
| $\lambda_2 = 0$ | 3.008 | 4.914 | 5.381 |
| $\lambda_2 = 1$ | 3.008 | 1.340 | 0.882 |
| $\lambda_2 = 2$ | 3.008 | 0.061 | 0.057 |
| $\lambda_2 = 4$ | 3.008 | 0.059 | 0.057 |
| $\lambda_2 = 8$ | 3.008 | 0.063 | 0.063 |

factor $C$, both OR-WDRO and our method are not sensitive to contamination factor $C$ in linear regression tasks.

Table 6: Excess risk with various contamination for linear regression.

| Contamination factor $C$ | Standard DRO | OR-WDRO [20] | UOT-DRO |
|---|---|---|---|
| 4 | 0.141 | 0.103 | 0.063 |
| 8 | 2.807 | 0.119 | 0.058 |
| 12 | 6.371 | 0.125 | 0.058 |

We use the same preprocessing step as in [20] to obtain a robust estimate of the expected value of the clean distribution, which involves a parameter $\hat{\varepsilon}$. This parameter can be viewed as an estimate of the proportion of data corruption. We evaluate the sensitivity of the model's performance to variations in $\hat{\varepsilon}$ and the result is presented in Table 7. We observe that our method achieves better robustness and is less sensitive to changes in $\hat{\varepsilon}$ compared to OR-WDRO in [20]. This is because OR-WDRO relies on $\hat{\varepsilon}$ during both the preprocessing and the solver optimization steps while our method only requires $\hat{\varepsilon}$ during the preprocessing step.

Table 7: Excess risk with various preprocessing parameter $\hat{\varepsilon}$ for linear regression ($\varepsilon = 0.1$).

| $\hat{\varepsilon}$ | standard DRO | OR-WDRO [20] | UOT-DRO |
|---|---|---|---|
| 0.05 | 2.661 | 0.088 | 0.049 |
| 0.1 | 2.661 | 0.066 | 0.049 |
| 0.2 | 2.661 | 0.064 | 0.050 |

## 7.7 Additional experiments on linear classification

We conducted additional experiments on linear classification, supplementing those presented in the main body of the paper. For these experments, we select $d = 10$. First, we evaluate the model performance across various sample sizes. As illustrated in Fig. 3, our method outperforms others in terms of both excess risk and accuracy. Next, we assess the model performance for different dimension $d$, as shown in Fig. 4. We observe that our method achieves superior robustness and maintains the accuracy around $94\%$.

In addition, we compare the computational efficiency of these methods in linear classification. As detailed in Table 8, OR-WDRO exhibits low computational time when the sample size is small, but it faces scalability issues as the dimension increases. In contrast, our method demonstrates stable computational times and superior performance across various sample sizes.

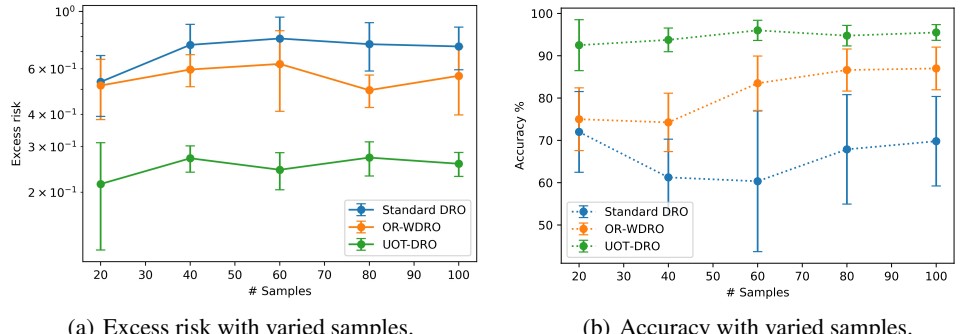

(a) Excess risk with varied samples.    (b) Accuracy with varied samples.

Figure 3: Excess risk and accuracy of standard DRO, OR-WDRO, and UOT-DRO with varied sample sizes for linear classification. The error bar denotes $\pm$ standard deviation.

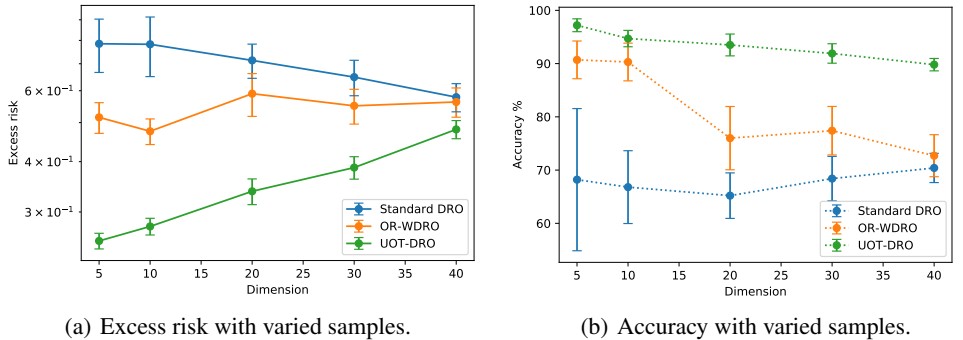

(a) Excess risk with varied samples.    (b) Accuracy with varied samples.

Figure 4: Excess risk and accuracy of standard DRO, OR-WDRO, and UOT-DRO with varied dimensions for linear classification. The error bar denotes $\pm$ standard deviation.

Table 8: Comparison of computation time, loss values, and accuracy for different methods for linear classification. 'ER' stands for excess risk.

| Sample | Standard DRO | | | OR-WDRO [20] | | | UOT-DRO | | |
|---|---|---|---|---|---|---|---|---|---|
| Size $n$ | Time | ER | Accuracy | Time | ER | Accuracy | Time | ER | Accuracy |
| 80 | 0.1 | 0.686 | 65% | 0.5 | 0.605 | 80% | 12.8 | 0.227 | 96% |
| 200 | 0.2 | 0.710 | 70% | 0.8 | 0.536 | 88% | 13.6 | 0.265 | 95% |
| 2000 | 3.1 | 1.013 | 43% | 28.8 | 0.546 | 96% | 13.5 | 0.299 | 98% |
| 5000 | 8.2 | 1.009 | 41% | 270.5 | 0.536 | 96% | 14.2 | 0.292 | 98% |
| 10000 | 19.4 | 1.016 | 30% | 1478.4 | 0.544 | 97% | 14.4 | 0.301 | 98% |
| 20000 | 64.2 | 1.021 | 35% | * | * | * | 14.9 | 0.295 | 99% |

As in the linear regression task, we evaluate the model's sensitivity to the preprocessing step, specifically to the parameter $\hat{\varepsilon}$. The parameter $\hat{\varepsilon}$ is again the estimate of the proportion of data corruption. As illustrated in Table 9, our method is less sensitive to the accuracy of the estimate $\hat{\varepsilon}$. The reason mirrors that in the linear regression task.

Table 9: Excess risk with various $\hat{\varepsilon}$ for linear classification.

| Parameter $\hat{\varepsilon}$ | Standard DRO | | OR-WDRO [20] | | UOT-DRO | |
|---|---|---|---|---|---|---|
| | Excess Risk | Accuracy | Excess Risk | Accuracy | Excess Risk | Accuracy |
| 0.05 | 0.713 | 67% | 0.447 | 81% | 0.283 | 93% |
| 0.1 | 0.713 | 67% | 0.505 | 85% | 0.283 | 93% |
| 0.2 | 0.713 | 67% | 0.484 | 86% | 0.282 | 93% |

Finally, we evaluate the sensitivity of the model performance to the tuning parameters $\lambda$ and $\beta$. We fix $C = 10$. Table 10 shows that our method achieves the accuracy above 90% across the range of $\beta$ values from 0.5 to 20 and $\lambda$ values from 5 to 20. This performance is better than that of standard DRO with accuracy 66%, and OR-WDRO with accuracy 79%. These simulation results indicate that our method is not sensitive to these parameters in linear classification tasks.

Table 10: Accuracy of various parameters for linear classification when $C = 10$. Standard DRO: 66%; OR-WDRO: 79%.

| Parameter $\beta$ | $\lambda = 1$ | $\lambda = 5$ | $\lambda = 10$ | $\lambda = 20$ | $\lambda = 40$ | $\lambda = 80$ |
|---|---|---|---|---|---|---|
| 0.1 | 65% | 71% | 77% | 88% | 94% | 94% |
| 0.5 | 81% | 90% | 95% | 95% | 95% | 95% |
| 2 | 91% | 94% | 94% | 94% | 93% | 93% |
| 8 | 93% | 93% | 92% | 92% | 90% | 85% |
| 20 | 92% | 92% | 91% | 90% | 82% | 68% |
| 40 | 91% | 89% | 85% | 75% | 63% | 57% |
| 100 | 93% | 86% | 74% | 61% | 54% | 47% |

## 7.8 Additional experiments on logistic regression

We evaluate the sensitivity of the model performance to the tuning parameters in logistic regression. As shown in Tables 11 and 12, our method achieves the accuracy above 85% across the considered ranges of $\lambda$ from 5 to 15 and $\beta$ from 0.4 to 10. Based on all of these simulation results, we find that our method's performance remains stable across a wide range of parameters.

Table 11: Loss and accuracy of various $\beta$ for logistic regression.

| Parameter $\beta$ | $\lambda = 5$ | | $\lambda = 10$ | | $\lambda = 15$ | |
|---|---|---|---|---|---|---|
| | Loss | Accuracy | Loss | Accuracy | Loss | Accuracy |
| 0.4 | 0.530 | 85% | 0.466 | 90% | 0.449 | 92% |
| 2 | 0.443 | 91% | 0.484 | 92% | 0.510 | 92% |
| 10 | 0.563 | 91% | 0.564 | 91% | 0.586 | 91% |

Table 12: Loss and accuracy of various $\lambda_2$ for logistic regression.

| Parameter $\lambda_2$ | $\lambda = 5$ | | $\lambda = 10$ | | $\lambda = 15$ | |
|---|---|---|---|---|---|---|
| | Loss | Accuracy | Loss | Accuracy | Loss | Accuracy |
| 0.1 | 0.269 | 93% | 0.360 | 89% | 0.431 | 87% |
| 0.4 | 0.388 | 92% | 0.451 | 91% | 0.476 | 91% |
| 1 | 0.440 | 94% | 0.478 | 93% | 0.509 | 94% |
| 4 | 0.613 | 90% | 0.601 | 91% | 0.600 | 91% |

