# OpenReview forum: "Outlier-Robust Distributionally Robust Optimization via Unbalanced Optimal Transport"
_NeurIPS.cc/2024/Conference — NeurIPS 2024 poster_

### Official Review · Reviewer_QxeQ · 2024-07-02

**Soundness:** 2
**Presentation:** 3
**Contribution:** 2
**Rating:** 5
**Confidence:** 2

**Summary:**

The paper proposes a new Distributionally Robust Optimization (DRO) framework based on Unbalanced Optimal Transport (UOT) distance. Under some conditions, the authors establish strong duality results for a Lagrangian penalty variation of the proposed problem. The designed algorithm is tested in linear, logistic regression and linear classification tasks where it shows an increased robustness to outliers in comparison to the considered baselines.

**Strengths:**

The paper proposes an original approach to the DRO problem utilizing the UOT distance to define the ambiguity set.  The authors provide theoretical justifications for the dual form of the problem under consideration and convergence analysis of the constructed algorithm. They empirically demonstrate that the proposed approach allows for increasing the robustness to outliers in comparison to the chosen baselines.

**Weaknesses:**

**Quality.** Some of the assumptions related to the ‘outlier-robustness’ property of the proposed approach are not sufficiently explained.

1) The authors claim that the ambiguity set of the proposed approach (UOT DRO) necessarily includes ‘the clean distribution $P^*$' (line 146-147) due to the soft marginal constraints in the UOT problem. However, actually, this ‘softness’ of the marginals depends on the chosen divergence $\mathcal{D}_{\phi_2}$ and the parameter $\beta$ in the unbalanced divergence definition (4). Typically, increasing the parameter $\beta$, one gets closer to the balanced OT formulation which does not have ‘outlier robustness’ property, and does not allow one to ignore the outliers in empirical dataset. This dependence on parameter $\beta$ is not explained and verified from theoretical or practical perspectives. Clarification of this point seems to be very important since this parameter is set differently in experiments considered in the paper.

2) Besides, the authors introduce the function $h$ penalizing potential distributions with outliers in an ambiguity set of the primal problem (6) of UOT-DRO. Thanks to this function, the solution (model) of the Lagrangian penalty reformulation of UOT-DRO will be less affected by outlier data as explained in lines 215-225. However, the authors provide explanations for the case of large $\lambda$, while in the experimental section they consider $\lambda=10$. The influence of outliers on the model in this case is still not obvious.

**Clarity.** Some parts of the text should be written more clearly, e.g., the formula in line 289. The choice of cost function and parameters in experiments (Section 5.1) should be explained.

**Evaluation.** The comparison with the baselines seems to be rather limited. The authors have cited several other works which propose approaches for the outlier-robust DRO problem [1,2,3,4]. However, the empirical evaluation of the proposed approach misses the comparison with some of these approaches [1,3,4].


**In summary**, I am not sufficiently convinced that the constructed approach actually solves the problem that it was designed for, i.e., allows for solving the DRO problem while effectively ignoring the outliers in the empirical dataset. The paper does not provide any theoretical results which support this claim. Thus, it should be accurately justified at least from the empirical side. However, the experimental part misses ablation study of parameters $\beta$, $\lambda$, and thorough comparison with other outlier-robust DRO approaches.

**Questions:**

1) Please explain the choice of cost function $c(\xi,\zeta)=|\xi-\zeta|$ in your experiments
2) How do you choose parameters $\beta$, $\lambda$ in your experiments? Is it an heuristic choice? An ablation study of these parameters is highly anticipated
3) Why is the performance of UOT-DRO for contamination $C=30$ higher than for $C=100$? How further increase of $C$ will influence the performance?
4) Why you did not perform comparison with additional outlier-robust DRO approaches [1,3,4] cited in the paper?

*Minor limitations*:
- bibliography is not in alphabetical order,
- line 286: $\theta_*$ is not defined,
- lines 311-312: '*only* requires *only* 20 seconds'.

**References.**

[1] Ramnath Kumar, Kushal Majmundar, Dheeraj Nagaraj, and Arun Sai Suggala. Stochastic re-weighted gradient descent via distributionally robust optimization. arXiv preprint arXiv:2306.09222, 2023.

[2] Sloan Nietert, Ziv Goldfeld, and Soroosh Shafiee. Outlier-robust Wasserstein dro. Advances in Neural Information Processing Systems, 36, 2024.

[3] Runtian Zhai, Chen Dan, Zico Kolter, and Pradeep Ravikumar. Doro: Distributional and outlier robust optimization. In International Conference on Machine Learning, pages 12345–12355. PMLR, 2021.

[4] R Tyrrell Rockafellar and Stanislav Uryasev. Optimization of conditional value-at-risk. Journal of Risk, 2:21–42, 2000.

**Limitations:**

The authors adequately addressed the limitations.

---

> ### Author Rebuttal · Authors · 2024-08-06
>
> We thank the reviewer for their thoughtful feedback. We address the concerns below.
>
> ---
> > **Quality 1: Chosen divergence and impact of $\beta$**
>
> Thank you for highlighting the chosen divergence in UOT. This work uses the well-studied KL divergence, which finds diverse applications in information theory, machine learning and optimal transport. Exploring other metrics or divergences is of course interesting and left as our future work.
>
> Thank you for highlighting the parameter $\beta$. We have added a thorough discussion on the impact of $\beta$ from both the theoretical and practical perspectives. The theoretical discussion can be found in **Common Response C2.3**. Practically, we conducted additional experiments on linear classification to assess the model's performance across a broader range of $\beta$. Since changes in $\beta$ affect the function values, we used accuracy as the performance measure. As shown in the new Table 3, our method achieves the accuracy above 90\% across the range of $\beta$ values from 0.5 to 20 and $\lambda$ values from 5 to 20. This performance is better than that of standard DRO with accuracy 66\%, and OR-WDRO with accuracy 79\%. Besides, as analyzed in **Common Response C2.3**, the model's performance degrades when $\beta$ is either too small or too large. Overall, these simulation results show that our method is relatively insensitive to the choice of $\beta$ and easy to be tuned.
>
> ---
> > **Quality 2: Explanation of our method and large $\lambda$**
>
> In lines 215-225, we explained our method using the case of large $\lambda$ because it simplifies the function, making it easier to explain. We would like to clarify that our method UOT-DRO has outlier robustness regardless of the $\lambda$ value. Please see **Common Response C1** for a general explanation.
>
> In Section 5.1 on linear regression, we show that Eq (15) can be simplified to Eq (16) if $\lambda\geq \lambda_2 + \kappa(\theta)$. This condition is satisfied when we select $\lambda=10$, $\lambda_2=5$, since the true $\theta$ we use to generate data has a norm $1$ and $\kappa(\theta) = \left\| [\theta,1]\right\|_*\leq 2$. We apologize for any lack of clarity previously. The influence of outliers is reduced through assigning a small weight thanks to the function $h$, see lines 215-225 and **Common Response C1** for more discussions.
>
> ---
> > **Clarity**
>
> Thank you for your suggestion regarding clarity. We will enhance the clarity in the revised draft. The cost functions for linear regression and classification are the same as those in [17] to facilitate fair comparison. The cost function of logistic regression is taken from the seminal work [15].
>
> ---
> > **Evaluation and Q4: Comparison with approaches in [1,3,4]**
>
> Through this comparison, our goal is to highlight the advantage of our method in handling outliers by utilizing unbalanced optimal transport. We did not compare our method with the approaches in [1] and [3] because the DRO problem they address differs from ours, making a direct comparison infeasible. Specifically, references [1] and [3] use the KL divergence to construct the ambiguity set, resulting in a KL DRO problem, whereas [2] and this paper employ Wasserstein distance, leading to a Wasserstein DRO problem. The comparison between this paper and [1] or [3] would be unfair since there are two key differences: the distance used to build the ambiguity set and method to deal with outliers. Any observed performance differences could not be solely attributed to the outlier-handling methods, making the comparison inconclusive. While it might be possible to adapt our unbalanced method for outlier handling to the KL DRO framework, this would constitute a different problem altogether, requiring a complete reevaluation, which we have left as future work due to time constraints. Regarding reference [4], it is a classical paper on CVaR that does not propose any method for handling outliers. Though the dual of CVaR can be seen as a DRO problem, it cannot be applied to handle outliers directly.
>
> ---
> > **Q1: Choice of cost function $c(\xi,\zeta)=\left\|\xi-\zeta\right\|_1$**
>
> We adopt the cost function for linear regression from [17] to ensure a fair comparison between our method and theirs.
>
> ---
> > **Q2: Parameter selection**
>
> The parameters are tuned based on the parameter selection guideline as outlined in **Common Response C2**. Moreover, we conducted a thorough sensitivity analysis in three tasks. As shown in Tables 3 and 4 for linear regression, Table 9 and new Table 3 for linear classification, new Tables 1 and 2 for logistic regression, our method performs well across a wide range of parameters and is not sensitive to the choice of parameters. Please see **Common Response C3** for a detailed discussion on sensitivity analysis.
>
> ---
> > **Q3: Performance of UOT-DRO for contamination $C=30$ and $C=100$**
>
> Recall that our method reduces the effect of outliers by assigning them a small (but non-zero) weight. When the contamination is high, the outlier points are significantly distant from the normal data. In this case, outlier points, despite their minimal weighting, may slightly influence model performance due to their extreme deviation from the normal data. To assess the impact of increasing $C$, we conducted additional experiments on linear classification, keeping the parameters fixed. The results indicate an accuracy of 95.8\% for $C=30$, 95.2\% for $C=100$, 95.0\% for $C=300$, 94.6\% for $C=600$, and 93.4\% for $C=1000$. Given that $C=1000$ represents a substantially large contamination level relative to the variance of the data distribution, which is $\sqrt{10}$, these findings suggest that our method is not sensitive to $C$ and performs well even at high contamination levels.
>
> ---
> > **Minor Limitations**
>
> Thank you for pointing out these typos. We have refined them in the updated draft.

---

> > ### Comment · Reviewer_QxeQ · 2024-08-14
> >
> > I thank the authors for additional clarifications and conducted experiments, and increase my score.

---

### Official Review · Reviewer_KUtP · 2024-07-07

**Soundness:** 2
**Presentation:** 3
**Contribution:** 2
**Rating:** 6
**Confidence:** 3

**Summary:**

This paper introduces a novel outlier-robust Wasserstein Distributionally Robust Optimization (WDRO) framework based on unbalanced optimal transport (UOT). For the UOT-DRO, the authors establish strong duality results under specific smoothness assumptions. To enhance computational efficiency, they propose a Lagrangian penalty variant of the problem and prove the strong duality of this variant as well. The authors then employ stochastic gradient descent (SGD) to solve the Lagrangian penalty variant, providing a theoretical convergence guarantee for their method.

**Strengths:**

1. This paper is well-written and easy to follow.

2. The paper introduces a novel formulation of DRO based on UOT to address outliers in the empirical distribution. Rather than adopting UOT directly, the authors consider a variant of UOT (introduced in equation (4)) that restricts the positive measure to be a probability measure, which aligns with the focus of DRO on probability measures. Additionally, the authors introduce a function $h$ to measure the distance to the uncontaminated distribution, effectively ruling out distributions close to the empirical distribution but containing more outliers.

3. The authors propose a Lagrangian penalty variant of the new outlier-robust DRO problem and employ stochastic gradient descent (SGD) to solve this variant with theoretical convergence guarantee for their method.

4. In experiments conducted on linear regression, logistic regression, and linear classification, the proposed algorithm demonstrates superior performance compared to the baseline methods WDRO and OR-DRO. This validates the efficiency and effectiveness of the proposed approach.

**Weaknesses:**

1. The current draft does not include a theoretical analysis of the excess risk associated with the proposed UOT-DRO. Including such an analysis would strengthen the paper by providing a deeper understanding of the performance and limitations of the method.
2. The discussion on parameter selection is currently limited. The UOT-DRO relies on several hyperparameters, including $\lambda$, $\lambda_2$, and $\beta$, which may not be entirely independent. Although Section 7.6 addresses hyperparameter selection for linear regression, the approach for selecting these parameters in other tasks remains unclear. A more comprehensive discussion on tuning these parameters for general cases would be beneficial.

**Questions:**

1. In line 238-239, the authors mentioned that outlier-robust WDRO is highly sensitive to $\varepsilon$, the contamination level in the outlier-robust optimal transport. However, proposition 12 in [A] provides a robustness guarantee when the chosen $\hat \varepsilon$ does not match the true $\varepsilon$. Also, the numerical experiments in [17] show that OR-WDRO can perform well when the the chosen $\hat \varepsilon$ does not match the true $\varepsilon$. Could you further explain why outlier-robust WDRO is sensitive to $\varepsilon$ in line 238-239?
2. It is encouraged to show an empirical evaluation on the convergence of the proposed Algorithm 1. This would help demonstrate the practical applicability and efficiency of the algorithm.
3. For the numerical experiments, could the authors further explain how the values of the hyperparameters were determined for linear classification and logistic regression? Additionally, the choice of hyperparameters for the baseline methods is also missing and should be included for a comprehensive comparison.

[A] Nietert, Sloan, Ziv Goldfeld, and Rachel Cummings. "Outlier-robust optimal transport: Duality, structure, and statistical analysis." *International Conference on Artificial Intelligence and Statistics*. PMLR, 2022.

**Limitations:**

The authors discuss the limitations of their work in Section 3. Notably, this work does not have a negative societal impact.

---

> ### Author Rebuttal · Authors · 2024-08-06
>
> We thank the reviewer for their thoughtful feedback. We address the concerns below.
>
> ---
> > **W1: Excess risk**
>
> Thank you for raising this point. Analyzing the excess risk of UOT-DRO is quite challenging due to the existence of unbalanced optimal transport. Moreover, while [17] gives the analysis of excess risk, the distance used in the excess risk analysis is different from the distance used to obtain the DRO tractable reformulation. Analyzing excess risk is of course interesting and left as our future work. This work mainly focuses on computational efficiency compared to previous methods.
>
> ---
> > **W2: Parameter selection**
>
> We agree that this warrants further discussion. We have added thorough discussions on the selection of all the parameters $\lambda,\beta,\lambda_2$. Please see **Common Responses C2 and C3**.
>
> ---
> > **Q1: Sensitivity to $\varepsilon$**
>
> Thank you for providing the additional interesting reference [A]. Upon careful review, we find that the analysis in [A] cannot be applied to assess the sensitivity to $\varepsilon$ in [17], since the distance that the reference [17] uses to define the DRO problem is different from that analyzed in [A]. The numerical experiments in [17] show that the performance of OR-WDRO deteriorates when $\hat{\varepsilon}$ is much smaller than the true $\varepsilon$, as illustrated in Figure 2 in [17]. This finding aligns with the simulation results presented in Tables 6 and 8 of our paper.
>
> The mechanism behind OR-WDRO is that it cuts off the leftmost $\hat{\varepsilon}$ percentile of the random value $\sup _ {\xi \in \Xi} \left\\{ l(\theta,\xi) - \lambda_1 \left\| \xi - \xi_0 \right\|^2- \lambda c(\xi,\zeta)\right\\}$ in Eq (12), which is induced by outliers, and then computes the average of the remaining $(1-\hat{\varepsilon})$ percentile. If $\hat{\varepsilon}$ is much smaller than the true $\varepsilon$, it results in inefficient outlier exclusion and thus poor performance of OR-WDRO. Therefore, OR-WDRO depends on a relatively accurate estimate of $\varepsilon$. In contrast, our unbalanced method does not rely on cutting off outliers but instead re-weights samples, offering a smoother approach. As a result, our method does not require an accurate estimate of $\varepsilon$. For more details about how our method deals with outliers, please see **Common Response C1**.
>
> ---
> > **Q2: Convergence of Algorithm 1**
>
> We run Algorithm 1 in linear regression and the convergence result is shown in the new Figure 1. Since $F^*$ is hard to compute, we plot the evolution of $F(\theta_t)$.
>
> ---
> > **Q3: Parameter tuning for the proposed method and the baseline method**
>
> Thank you for raising this significant point. The guidelines for parameter selections can be found in **Common Response C2**. Following the guidelines, we tune the parameters and we find that the performance is not sensitive to parameter selections in three tasks. For more details about sensitivity analysis, please see **Common Response C3** and the attached PDF file, where we conducted additional experiments on linear classification and logistic regression.
>
> The baseline method proposed in [17] requires tuning the parameters $\lambda,\sigma,\epsilon$. We adopted the parameter selections from the original implementation of [17], which we find is nearly optimally tuned. Specifically, it sets $\sigma = \sqrt{d}$, see discussions in [17]; $\epsilon=0.1$, aligning with the 10\% data corruption rate; $\rho = 0.1$, corresponding to the size of Wasserstein perturbation. We will include these in the updated draft.

---

> > ### Comment · Reviewer_KUtP · 2024-08-13
> >
> > I thank the authors for answering my questions. I will keep my score.

---

> > > ### Author Response · Authors · 2024-08-13
> > >
> > > We thank the reviewer for the kind response!

---

### Official Review · Reviewer_8Mev · 2024-07-13

**Soundness:** 3
**Presentation:** 3
**Contribution:** 3
**Rating:** 7
**Confidence:** 5

**Summary:**

The authors leverage unbalanced optimal transport (UOT) to build a new DRO model. Strong dual reformulations together with efficient algorithm design have been proposed. Numerical studies on convex loss function demonstrate the superior performance of this framework.

**Strengths:**

Overall this is a good paper. The idea for using UOT is innovative and useful. Theoretical contributions are solid and I have checked their proofs and to the best of my knowledge, I don’t find mistakes. I leave some comments in next section, which may further help the authors improve the quality of this paper in terms of literature review and theoretical soundness.

**Weaknesses:**

1. I think the authors may miss an important citation and comparison:
    [JGX2023] Wang J, Gao R, Xie Y (2023) Sinkhorn distributionally robust optimization. arXiv preprint arXiv:2109.11926.

Specifically, the authors considered UOT-based DRO, where UOT is a variant of entropic regularized OT distance. In contrast, [JGX23] also considered another variant of entropic regularized OT distance (which they call it as Sinkhorn distance) to construct the ambiguity set:

$$S(\mathbb{P}, \widehat{\mathbb{P}}) = \inf_{\gamma \in \Gamma(\mathbb{P}, \widehat{\mathbb{P}})}~\Big\{
\mathbb{E}_{(\xi, \zeta)\sim \gamma}[c(\xi,\zeta)] + \beta \mathbb{E}_{(\xi, \zeta)\sim \gamma}\left[
\log\frac{\mathrm{d}\gamma(\xi,\zeta)}{\mathrm{d}\nu(\xi)\mathrm{d}\widehat{\mathbb{P}}(\zeta)}
\right]
\Big\}$$

where  $\nu(\cdot)$ can be any reference measure (such as Lebesgue measure) on $\Xi$. The difference is that the authors in this paper considered relative entropy regularization between  $\bar{\mathbb{P}}$ and $\widehat{\mathbb{P}}$, and [JGX23] considered relative entropy regularization between conditional conditional distribution $\gamma_{\zeta}(\cdot)$  and measure  $\nu(\cdot)$. Also, please compare the dual reformulation in this paper and that in [JGX23]. The expressions share some connections.


In addition, the usage of entropic regularization for Wasserstein DRO has been largely explored in other literature. I hope the authors also add the comparisons among them.

    [JY2022] J. Wang and Y. Xie, “A data-driven approach to robust hypothesis testing using sinkhorn uncertainty sets,” in 2022 IEEE International Symposium on Information Theory (ISIT). IEEE, 2022, pp. 3315–3320.

     [WFJ2023] 	W.Azizian, F.Iutzeler, and J.Malick “Regularization for wasserstein distributionally robust optimization,” ESAIM:Control, Optimisation and Calculus of Variations, vol. 29, p. 33, 2023.

     [JRY2022] J. Wang, R. Moore, Y. Xie, and R. Kamaleswaran, “Improving sepsis prediction model generalization with optimal transport,” in Machine Learning for Health. PMLR, 2022, pp. 474–488

     [SZ2023] 	S.-B. Yang and Z. Li, “Distributionally robust chance-constrained optimization with sinkhorn ambiguity set,” AIChE Journal, vol. 69, no. 10, p. e18177, 2023.

     [CFAB2023] C. Dapogny, F. Iutzeler, A. Meda, and B. Thibert, “Entropy-regularized wasserstein distributionally robust shape and topology optimization,” Structural and Multidisciplinary Optimization, vol. 66, no. 3, p. 42, 2023.

     [JGX2024] Wang J, Gao R, Xie Y (2024) Non-Convex Robust Hypothesis Testing using Sinkhorn Uncertainty sets. in 2024 IEEE International Symposium on Information Theory (ISIT). IEEE, 2024

**Questions:**

2. In Theorem 1, the assumption seems too restrictive, including boundedness of $f_{\lambda}(\theta,\zeta)$, differentiability and concavity of $L(\theta,\xi)$, differentiability and concavity of $c$, and strict positivity of dual variable $\lambda^*$. In existing DRO literature such as [JGX2023], only the finiteness of cost function, lighted tailed condition, measurability of loss, and law of total probability is required. In standard WDRO, the technical assumption for strong duality holds is mild as well.

3. I also believe the assumption that dual variable is strictly positive is not necessary: when you take sufficiently large radius, the constraint is unbinding and the dual variable $\lambda^*=0$. Please follow existing DRO literature to explore how to build strong duality in this case.

4. In line 220, the authors implicitly used the assumption that $c(\zeta,\zeta)=0$. I don’t remember whether it has been assumed in the main context before.

5. I have concerns regarding Theorem 3. The algorithm design is actually inspired from [Sinha et. Al 2017]. The authors in that reference studied optimization for non convex loss function, such as neural networks, and perform biased SGD update to achieve near-stationary points. However, the authors in this paper only considered convex optimization. I strongly believe their results can be extended for non-convex smooth optimization. Please consider the exploration in this direction. Some useful references include

[HCH2021] Hu Y, Chen X, He N (2021) On the bias-variance-cost tradeoff of stochastic optimization. Advances in Neural Information Processing Systems.

[ HZCH2020] Hu Y, Zhang S, Chen X, He N (2020) Biased stochastic first-order methods for conditional stochas- tic optimization and applications in meta learning. Advances in Neural Information Processing Systems, volume 33, 2759–2770.


6. For the numerical study part, the exploration of outlier DRO with neural network loss functions is also quite promising.

---

> ### Author Rebuttal · Authors · 2024-08-06
>
> We thank the reviewer for your thoughtful feedback. Please see the detailed response below.
>
> ---
> > **W1(Q1): Reference [JGX2023]**
>
> We thank the reviewer for bringing our attention to the reference [JGX2023] and other related references about regularized optimal transport distance. We will add these references and additional comparative analyses into our revised manuscript.
>
> The fundamental difference between regularized and unbalanced optimal transport distance lies in whether the marginal distributions of the couplings are allowed to be different from the empirical distribution. Take Sinkhorn Wasserstein DRO (WDRO) in [JGX23] as an example. Both Sinkhorn WDRO and unbalanced WDRO use the KL divergence, albeit in fundamentally different ways. In Sinkhorn WDRO, the KL divergence acts as a regularization term that facilitates computational efficiency at the expense of computation error. Importantly, the coupling $\gamma \in \Gamma(\hat{\mathbb{P}},\mathbb{P})$ in Sinkhorn WDRO must have at least one marginal distribution that is identical to the empirical distribution $\hat{\mathbb{P}}$. In contrast, unbalanced WDRO employs KL divergence to penalize discrepancies between marginal distributions, allowing for the coupling $\gamma\in \Gamma(\bar{\mathbb{P}},\mathbb{P})$ to have a different marginal distribution $\bar{\mathbb{P}}$. One of the resulting technical differences, as the reviewer pointed out, is the relative entropy regularization between two couples of distributions.
>
> The strong duality of Lagrangian penalty formulation of several related problems is as follows:
>
> Traditional WDRO: $\mathbb{E}_{\zeta \sim \hat{\mathbb{P}}} \Big[ \sup _ {\xi}  \\{  L(\xi)-\lambda c(\xi,\zeta) \\} \Big],$
>
> Sinkhorn WDRO: $\lambda \beta \mathbb{E} _ {\zeta\sim \hat{\mathbb{P}}} \Big[ \log  \mathbb{E} _ {\xi\sim \nu } \Big[{\rm{exp}} \left(\frac{L(\xi)-\lambda c(\xi,\zeta)}{\lambda\beta} \right) \Big]\Big],$
>
> Unbalanced WDRO: $\lambda \beta \log \mathbb{E} _ {\zeta\sim \hat{\mathbb{P}}} \Big[ {\rm{exp}}\left( \frac{\sup _ {\xi} \\{L(\xi)-\lambda c(\xi,\zeta)\\}} {\lambda\beta}\right)\Big].$
>
> From these dual formulations, Sinkhorn WDRO modifies traditional WDRO by substituting the $\sup$ with a log-sum-exp smoothing function, thereby enhance the computational efficiency. In contrast, unbalanced WDRO employs the log-sum-exp function differently. Here, it is used to re-weight samples from $\hat{\mathbb{P}}$ rather than replacing the $\sup$ operation. As a result, outliers exert a reduced impact on the decision-making process. Computationally, unbalanced is more challenging as it involves an inner-maximization problem, similar to traditional WDRO.
>
> ---
> > **Q2: Restrictive assumptions in Theorem 1**
>
> Compared to Sinkhorn DRO in [JGX2023], the additional assumption about the differentiability and concavity of $L(\theta,\zeta)$ is due to the existence of an inner sup in the dual formulation, which is smoothed out in Sinkhorn DRO. Another assumption about $\lambda^*=0$ is discussed in Q3. Relaxing these assumptions is left as our future work.
>
> ---
> > **Q3: Strictly positive dual variable is not necessary**
>
> It may happen that $\lambda^*=0$ when we select a sufficiently large radius. However, this scenario is uncommon in practice because choosing a very large radius typically results in overly conservative outcomes that are generally less appealing. We agree that removing this assumption is promising under additional conditions. We plan to address this in our future work as it is not straightforward by applying techniques from current DRO literature, e.g., [JGX2023].
>
> ---
> > **Q4: Implicit assumption $c(\zeta,\zeta)=0$**
>
> Thank you for raising this point. We have added it in the updated version.
>
> ---
> > **Q5: Extension to non-convex smooth optimization**
>
> Thank you very much for providing these two related references. We will incorporated them in our revised draft. Thank you for proposing the non-convex smooth optimization problem. This is definitely interesting and left as our future work.
>
> ---
> > **Q6: Neural network implementation**
>
> Thank you for affirming the potential of our method for use with neural networks (NNs). We believe our method can be applied to NN loss functions, and we plan to explore this in our future work.

---

> > ### Comment · Reviewer_8Mev · 2024-08-13
> >
> > I thank the authors for successfully addressing all my concerns. I will raise my score to 7

---

> > > ### Author Response · Authors · 2024-08-13
> > >
> > > We thank the reviewer for the kind response and for increasing the score.

---

### Author Rebuttal · Authors · 2024-08-06

# Common Response

> **C1: How does our method deal with outliers?**

Consider the Lagrangian penalty problem of minimizing $F(\theta) = \sum_{i=1}^n \exp\left(\frac{f _ {\lambda}(\theta,\hat{\zeta}_i)}  {\lambda \beta} \right)$ in Eq (13), where $\hat{\zeta}_i$ is the $i$-th sample that may be an outlier. The objective function $F$ comprises $n$ terms, where each term depends on the sample $\hat{\zeta}_i$, respectively. The key to achieving outlier robustness against outliers lies in reducing their contributions to the function $F$ compared to the normal data. The function $h$ helps this by ensuring that $f _ {\lambda}$ outputs a minimal value at outlier points regardless of the $\lambda$ value. Specifically, for an outlier $\hat{\zeta}_i$, the term  $\exp\left(\frac{f _ {\lambda}(\theta,\hat{\zeta}_i)}  {\lambda \beta} \right)$ would be very small and contribute less to the overall objective function $F$.

---
> **C2: Parameter selections.**

Here, we discuss some guidelines for selecting the parameters $\lambda,\lambda_2,\beta$ in Algorithm 1.

**C2.1: $\lambda$.** The parameter $\lambda$ is commonly used in DRO literature [25] as a penalty coefficient. A larger value of $\lambda$ makes the DRO problem get close to the empirical risk minimization, resulting in a model that performs well on the empirical distribution but is less robust to Wasserstein perturbations.

**C2.2: $\lambda_2$.** The parameter $\lambda_2$ represents the credibility level assigned to the function $h$.  A larger value of $\lambda_2$ should be selected when there is high confidence in the reliability of $h$, meaning that $h(\xi)$ is very likely to yield a large value at outlier points. Conversely, if the prior knowledge provided by $h$ is considered unreliable, the value of $\lambda_2$ should be reduced. If there is no reliable prior knowledge at all, in which we should select $\lambda_2=0$, achieving good performance is impossible as discussed in the robust statistics literature [18].

**C2.3 : $\beta$.** The parameter $\beta$ quantifies the degree of penalization for the mismatch of marginal distributions. A larger value of $\beta$ places more emphasis on minimizing this mismatch, possibly at the expense of increasing the transportation cost, thereby making the unbalanced optimal transport distance close to the balanced one. Conversely, a small value of $\beta$ allows for greater flexibility in the distribution mismatch, which can enhance the model's robustness to outliers. However, when $\beta$ is extremely small, the mismatch incurs little penalty, and the computed distance may fail to accurately represent the true distance between the distributions.  In this case, the resulting DRO problem would include too many unlikely distributions in the ambiguity set, leading to an overly conservative model.

Theoretically, we discuss the impact of parameter $\beta$ from the perspective of original problem formulation (13). Following the discussion in Common Response C1, consider the objective function $F(\theta) = \sum_{i=1}^n \exp\left(\frac{f _ {\lambda}(\theta,\hat{\zeta}_i)}  {\lambda \beta} \right)$, which consists $n$ terms, each induced by a data point. A suitable value of $\beta$ would enable $\exp\left(\frac{f _ {\lambda}(\theta,\hat{\zeta}_i)}  {\lambda \beta} \right)$ to be smaller at an outlier point compared to normal data, thus reducing the impact of outliers. However, if $\beta$ is overly large, each term will equally approach 1 regardless of the value of $f _ {\lambda}$, thus failing to reduce the impact of outliers. This observation aligns with the previous analysis that a large value of $\beta$ makes the unbalanced method lose the outlier robustness as it approaches the balanced one. Conversely, an extremely small $\beta$ can also degrade performance because the objective function becomes overly focused on the sample that yields the largest value of $f _ {\lambda}$, neglecting the rest.

---
**C3: Parameter sensitivity.** When tuning the parameters, we find that our method allows a wide range of parameter selections without significant performance variations in linear regression, classification and logistic regression problems. This is verified through a comprehensive sensitivity evaluation. For linear regression, as shown in Tables 3 and 4, the model's performance shows minimal sensitivity to variations in these parameters. For linear classification, as shown in Table 9 and new Table 3, our method maintains an accuracy above 90\% across a variety of parameter settings. For logistic regression, we conducted additional experiments to evaluate the sensitivity of parameters $\lambda,\beta,\lambda_2$. As shown in the new Tables 1 and 2, our method achieves the accuracy above 90\% in most cases across the considered ranges of $\lambda$ from 5 to 15 and $\beta$ from 0.4 to 10. Based on all of these simulation results, we find that our method's performance remains stable across a wide range of parameters and it is easy to tune these parameters.

---

### Comment · Area_Chair_23PQ · 2024-08-12

Dear Reviewers,

We encourage you to participate in the discussion actively. Please post a comment on the authors' response, either to highlight any remaining concerns, acknowledge that some of your concerns have been addressed, or simply to confirm that you have reviewed their response.

Best wishes, AC

---

### Decision · Program_Chairs · 2024-09-25

**Decision:**

Accept (poster)

**Comment:**

The authors present a novel approach by leveraging unbalanced optimal transport (UOT) to develop a new Wasserstein Distributionally Robust Optimization (WDRO) framework, accompanied by strong duality results for a Lagrangian penalty variation of the proposed problem. The framework is tested across multiple scenarios, including linear, logistic regression, and linear classification tasks, demonstrating enhanced robustness to outliers. Although there are areas for further improvement, such as deeper theoretical analysis and sensitivity to contamination levels, the paper offers valuable contributions and has sufficient merit for acceptance.